# Evaluation and Bias Correction of Probabilistic Volcanic Ash Forecasts

Alice Crawford[1], Tianfeng Chai[1,2], Binyu Wang[3,4], Allison Ring[5], Barbara Stunder[1], Christopher P. Loughner[1], Michael Pavolonis[6], and Justin Sieglaff[7]

[1]NOAA Air Resources Laboratory, College Park, MD, USA
[2]Cooperative Institute for Satellite and Earth System Studies (CISESS), University of Maryland, College Park, MD, USA
[3]NOAA National Centers for Environmental Prediction, Environmental Modeling Center, College Park, MD, USA
[4]IM Systems Group, Rockville, MD, USA
[5]Department of Atmospheric and Ocean Science, University of Maryland, College Park, MD, USA
[6]NOAA National Environmental Satellite, Data and Information Service (NESDIS), Madison, WI, USA
[7]Cooperative Institute for Meteorological Satellite Studies (CIMSS), Madison, WI, USA

**Correspondence:** Alice Crawford (alice.crawford@noaa.gov)

**Abstract.** Satellite retrievals of column mass loading of volcanic ash are incorporated into the HYSPLIT transport and dispersion modeling system for source determination, bias correction, and forecast verification of probabilistic ash forecasts of a short eruption of Bezymianny in Kamchatka. The probabilistic forecasts are generated with a dispersion model ensemble created by driving HYSPLIT with 31 members of the NOAA global ensemble forecast system (GEFS). An inversion algorithm is used for source determination. A bias correction procedure called cumulative distribution function (CDF) matching is used to very effectively reduce bias. Evaluation is performed with rank histograms, reliability diagrams, fractions skill score, and precision recall curves. Particular attention is paid to forecasting the end of life of the ash cloud when only small areas are still detectable in satellite imagery. We find indications that the simulated dispersion of the ash cloud does not represent the observed dispersion well, resulting in difficulty simulating the observed evolution of the ash cloud area. This can be ameliorated with the bias correction procedure. Individual model runs struggle to capture the exact placement and shape of the small areas of ash left near the end of the clouds lifetime. The ensemble tends to be overconfident, but does capture the range of possibilities of ash cloud placement. Probabilistic forecasts such as ensemble relative frequency of exceedance and agreement in percentile levels are suited for strategies in which areas with certain concentrations or column mass loadings of ash need to be avoided with a chosen amount of confidence.

## 1 Introduction

We describe modeling efforts to provide quantitative probabilistic forecasts of concentrations of volcanic ash for use by the aviation sector with an emphasis on forecast verification in which the properties of the joint distribution of forecasts and observations are explored through the use of various statistical measures. No standard set of verification metrics for probabilistic volcanic ash forecasts is currently in use.

Currently, forecasts for ash issued by Volcanic Ash Advisory Centers, VAACs, consist of polygons denoting the area of discernible ash. However, within the next five years, VAACs may be producing gridded products of ensemble relative frequency of exceedances of prescribed concentration thresholds  (International Civil Aviation Organization Meteorology Panel, 2019).

Verification and evaluation metrics for probabilistic forecasts of a set of discrete predictands will guide model development and the construction of dispersion model ensembles and provide valuable information to end users of the forecasts. For instance,
Prata et al. (2019) have developed a risk-matrix approach for a risk-based approach to flight planning. The approach depends on identifying probability thresholds for less likely, likely and very likely events. For a proof of concept demonstration  Prata et al. (2019) used a mostly ad hoc set of probability thresholds for the events. Here we show forecast verification metrics that can aid in producing meaningful and appropriate probability thresholds for use in risk-based approaches.

Ash forecasts often have a large bias largely due to uncertainties in the source term. For binary predictands (e.g. forecasts
which indicate areas of ash above a given threshold), a frequency bias leads to either over or under prediction of the extent of the above threshold area. We discuss measures of bias and use a bias correction procedure called cumulative distribution function, CDF, matching to successfully reduce bias.

Sect. 2 describes the eruption of Bezymianny in October of 2020 which will be used as a case study. Sect. 3 discusses the observations of ash mass column loading. These observations are used to help define the source term for input into the model
through both an inversion algorithm, Sect. 4.3, and a bias correction procedure, Sect. 5. They are also utilized for evaluation of probabilistic forecasts in Sect. 6.

## 2   Description of eruption of Bezymianny

Bezymianny began erupting on 21 October 2020 shortly after 20:00 UTC. Initial estimates reported the ash cloud reached a plume height of around 9 km  (Sennert, 2020). Here and throughout the text all heights are in reference to mean sea level. The
main ash producing portion of the eruption lasted less than two hours, although emissions of gas, steam and possibly some ash continued to be observed until around 03:00 UTC the next day  (Sennert, 2020). Later analysis of the eruption column height by Horváth et al. (2021) using geometric estimation gave an initial height of around 9 km at 20:30 UTC, a top height of about 13 km at 20:40 UTC, and a maximum altitude of an overshooting plume top of 15.3 to 15.7 km by 20:50 UTC.

Meteorological conditions above the vent are discussed in Appendix A. Wind speeds reached about 14 m s$^{-1}$ at 350 mb
according to numerical weather prediction models.

The eruption makes a particularly good test case for a few reasons. The emission is relatively short and uncomplicated and thus the source term has less uncertainty than in longer duration eruptions which may consist of many emissions of varying intensities and eruption column heights. The resulting ash cloud forms a complicated three dimensional structure as it is stretched and folded by the wind field over the course of less than a day. As shown later, the exact location and shape of these
structures is difficult to forecast.

In many larger eruptions, such as Kastaochi (Crawford et al., 2016) and Raikoke (de Leeuw et al., 2021), most of the ash is drawn toward and around an area of low pressure, the location of which is fairly easy to forecast. The end fate of these large

ash clouds is similar to that of the Bezymianny cloud discussed here. However, it takes longer to occur and the location of the smaller structures can be even more difficult to forecast because of the time passed.

The short duration of this Bezymianny eruption makes running and testing many different simulation setups more tractable. Most importantly perhaps is that eruptions of this size occur fairly frequently. One aspect of concern to aviation which has not received much attention in the literature is how well forecasts predict the dissipation of the ash cloud by which we mean its gradual disappearance. This question becomes more urgent as areas of detectable ash become larger due to improved detection methods and there may be more reliance on models to estimate when concentrations drop below a certain threshold.

## 3 Satellite data

Satellite retrievals were produced as part of the Volcanic Cloud Analysis Toolkit (VOLCAT). For this eruption, satellite retrievals are every 10 minutes from 21 October 2020, 20:40 UTC to 22 October 2020, 21:10 UTC. The data was from the Himarawi-8 Advanced Himawari Imager, AHI, Fulldisk scans. Satellite retrievals provide information on atmospheric column mass loading, hereafter just referred to as mass loading, cloud top height, and effective radius. The mass loading is the field
which is most utilized here.

  The VOLCAT ash detection algorithm (Pavolonis et al., 2015a, b) is designed to mimic human expert analysis of multispectral satellite imagery. Visual comparison to the corresponding satellite imagery indicates that the VOLCAT algorithm accurately captured the spatial extent of the Bezy cloud throughout the analysis period, with a scene dependent lower limit of detection of approximately 0.2 mg m$^{-2}$ (Prata and Prata, 2012). The mass loading is retrieved using the method described
in Pavolonis et al. (2013). Ash grains are assumed to follow a lognormal distribution. The satellite retrievals are based on measurements from the 10-13 $\mu$m portion of the EM spectrum, so the retrieval will be sensitive to ash grains with a radius of approximately 1 $\mu$m or greater. Ash is assumed to be spherical and the refractive index of andesite is used (Pavolonis et al., 2013).

  Figure 1 shows time series of some relevant properties of the satellite retrievals. The total column mass is retrieved for each
satellite pixel where ash is detected. The total mass is computed by summing the product of column ash loading and pixel area across all pixels in the detected ash cloud. The total mass increases quickly at an approximately constant rate for the first hour (10/21 20:40 through 21:40 UTC). The total area of the cloud keeps increasing sharply for about another two hours.

  A rough calculation of mass eruption rate, MER, from the change in mass in the satellite retrievals during this time (Figure 1(a)) gives approximately $1.5 \times 10^4$ kg s$^{-1}$. After the first hour, the total mass decreases and the rate of decrease is well
described by an exponential decay with a half life of about 4 hours. Presumably the decrease in detected mass over time is due to physical processes such as dilution below detection limits, gravitational settling, and wet and dry deposition.

  Maximum retrieved heights are around 10 km during the initial eruption period and increase to around 12 km. It is expected that these heights would be somewhat lower than those from the geometric estimation in Horváth et al. (2021). The retrieved heights are not utilized here for the inversion algorithm or the evaluation although they will be considered in future work.

For comparison to model output, the satellite data is parallax corrected using estimated cloud top heights and then composited by first regridding to a regular 0.1 degree latitude-longitude grid and subsequently taking the average of all retrievals within an hour time frame. Cumulative distribution functions, CDF, of the composited data are shown in Figure 2. Mass loading values decrease over time as might be expected. Only the first two time periods contain mass loading values greater than 10 g m$^{-2}$.

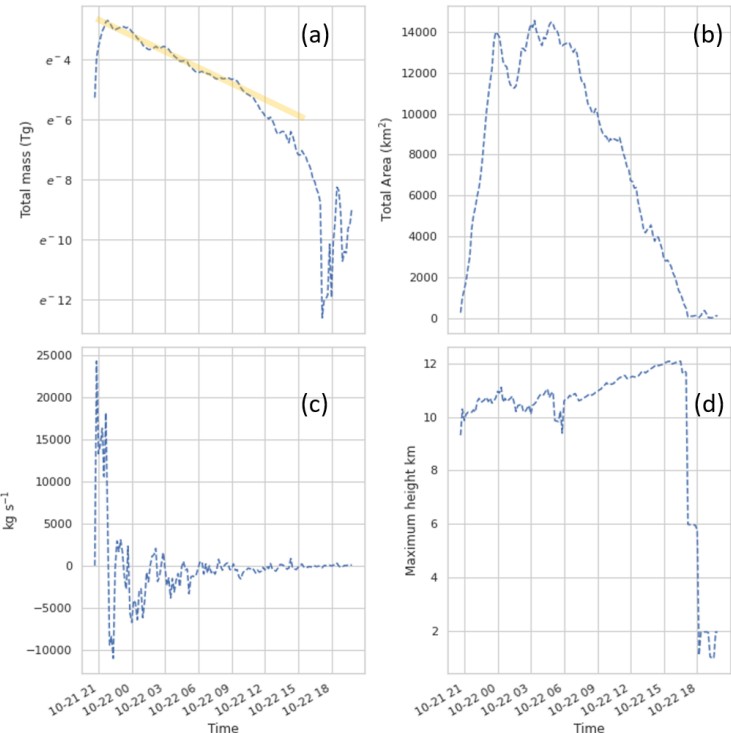

**Figure 1.** (a) Total mass in satellite retrieval as a function of time. The y axis is a natural log scale and the orange line indicates a period of exponential decay. (b) Total area of ash in satellite retrieval. (c) Rate of change of mass in satellite retrieval calculated by differencing the mass between subsequent satellite retrievals and dividing by the time between them. (d) Maximum height in satellite retrieval.

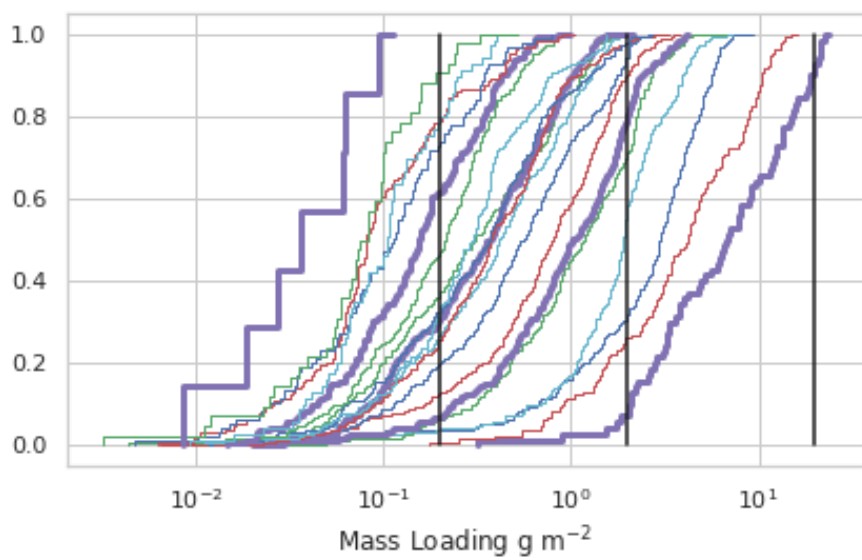

**Figure 2.** Cumulative distribution functions of composited VOLCAT retrievals. The dark purple line on the right is for satellite data from 21 October 2020 21–22 UTC. Colors repeat every 5 hours. The dark purple line on the left is for satellite data from 22 October 2020 17–18 UTC. The vertical black lines denote 0.2, 2, and 20 g m$^{-2}$

## 4   HYSPLIT transport and dispersion model

HYSPLIT is a widely used Lagrangian atmospheric transport and dispersion model, ATDM, developed and maintained by the Air Resources Laboratory, ARL, at NOAA  (Stein et al., 2015a). It is used operationally by the Washington and Anchorage VAACs to aid in producing Volcanic Ash Advisories, VAAs. HYSPLIT requires both meteorological data and a source term as input. The source term describes when and where mass is initialized in the model.

HYSPLIT was run as a Lagrangian particle model and choice of computational particle number is discussed more in Ap-
pendix C. For all model runs the horizontal output resolution was $0.1^o \times 0.1^o$. This is smaller than in many other studies. For instance  Harvey et al. (2020) uses a $0.375^o \times 0.5625^o$ grid for mass loading.  Kristiansen et al. (2012) uses a $0.25^o \times 0.25^o$ grid. However more recently  Folch et al. (2022) uses a $0.05^o \times 0.05^o$ grid. A higher horizontal resolution was chosen partly because the observed ash cloud was stretched into quite a thin line of ash, the properties of which are not captured well by a coarse resolution grid. Also, coarser grid sizes can be investigated simply by averaging over the finer one.

The time for different particle sizes to produce significantly different ash cloud distributions was tested using the method described in  Crawford (2020). By this method, it is determined that particles with diameters 6 $\mu$m and 0.6 $\mu$m do not separate throughout the length of the period of interest. However, 20 $\mu$m particles start to separate at about 22 October 2020 00 UTC. This is discussed in more detail in Appendix B. Most runs were performed with a single particle size of 6 $\mu$m. However, some runs were performed with a 20 $\mu$m or 50 $\mu$m particle size.

Table 1 summarizes the runs discussed in this paper. RunA2, RunC, RunD, and RunE are discussed only briefly to provide some information on the effect of varying particle size and initial eruption column width. RunB was used to investigate the effect of utilizing different observational time periods in the inversion algorithm as discussed in Sect. 4.3. RunA and RunM were used for the bias correction procedure and evaluation in Sect. 5.1, 5.2, and 6.

**Table 1.** Table describing HYSPLIT runs.

| RunID | Source Diameter | Particle | Source term | Met |
|-------|-----------------|----------|-------------|-----|
| RunA  | 1 km  | 6$\mu$m  | 2h emission | GEFS |
| RunA2 | 20 km | 6$\mu$m  | 2h emission | GEFS |
| RunB  | 1 km  | 6$\mu$m  | inversion   | GFS $0.25^o$ |
| RunC  | 20 km | 6$\mu$m  | inversion   | GFS $0.25^o$ |
| RunD  | 1 km  | 20$\mu$m | inversion   | GFS $0.25^o$ |
| RunE  | 1 km  | 50$\mu$m | inversion   | GFS $0.25^o$ |
| RunM  | 1 km  | 6$\mu$m  | inversion   | GEFS |

## 4.1 Meteorological models

HYSPLIT utilizes wind fields and other information from a numerical weather prediction, NWP, model as inputs (Stein et al., 2015a). For this study we considered two NWP models produced by National Centers for Environmental Prediction (NCEP), the Global Forecasting System (GFS) $0.25^o$ resolution and the Global Ensemble Forecast System, GEFS. The GEFS has 31 members and $0.5^o$ horizontal resolution. The GEFS has 23 vertical pressure levels. Levels from 1000 hPa to 900 hPa are spaced 25 hPa apart and higher levels are spaced 50 hPa apart, with highest levels at 50 and then 20 hPa. For this particular dataset the 150 mb level was not utilized because of an error in converting the grib files to HYSPLIT input format. The GFS $0.25^o$ has 54 pressure-sigma hybrid levels. Both datasets have 3 h temporal resolution and both forecasts are produced every 6 hours. For the HYSPLIT runs, the most recent forecast was always utilized, e.g. for a 12 hour forecast two or three forecast cycles would be utilized. This does not faithfully represent an operational framework in which future forecast cycles would obviously not be available. We expect it to produce a better forecast than for instance using only one cycle because of increased accuracy in wind speeds and directions. However this has not been investigated in depth and some inconsistency in the wind fields may be introduced which could result in degraded model performance. Meteorological conditions are discussed in more detail in Appendix A.

## 4.2 Source term for RunA

RunA is a control run. The source term was initially estimated using methods similar to those currently employed in an operational setting. The start time of the eruption, duration, eruption column height, and eruption column width were all determined by human interpretation of available observations. Emission start was 21 October 2020 at 20 UTC. Emission duration was 2 h. Initial mass distribution was uniform throughout a cylinder centered at the vent with width 1 km. The base of the cylinder was at 2.88 km, the vent height. The top of the cylinder was at 12.88 km. This is similar to current default operational settings at Washington and some other VAACs (Witham et al., 2007; Beckett et al., 2020). A uniform vertical mass distribution is not particularly realistic, but it is practical for current operations as it provides information on movement of ash from all heights that can then be further interpreted by the analyst issuing the final forecast.

Initially the constant emission rate was set at $1.5 \times 10^4$ kg s$^{-1}$ as this was consistent with a rough estimate from Figure 1(c). However it was revised to approximately $3.75 \times 10^3$ kg s$^{-1}$ as described in Sect. 5.

The emission rate in the model is representative of the mass eruption rate of fine ash, MER$_f$. For comparison, the widely utilized empirical relationship between total mass eruption rate, MER, and eruption column height given in Mastin et al. (2009) and a mass fraction of fine ash of 0.1 would result in an MER$_f$ of $4.85 \times 10^4$ kg s$^{-1}$ for a 10 km high eruption column. This estimate is about an order of magnitude larger than the value we use, but the uncertainty in it is at least that large given that fraction of fine ash can vary between about 0.01 and 0.5 (Mastin et al., 2009), the eruption column height estimate has an uncertainty of a few kilometers, and the the empirical fit itself has a large uncertainty (Mastin et al., 2009).

## 4.3 Inversion algorithm

The inversion algorithm described in  Chai et al. (2015, 2017) was used to construct source terms for RunB and RunM. In short, forward HYSPLIT dispersion runs were performed to construct a transfer coefficient matrix (**TCM**) which provides the contribution of each possible source location to each measurement. The unknown emissions are estimated by minimizing a cost functional that integrates the differences between the model predictions and the observations and deviations of the solution from an a priori  (Chai et al., 2017). As prior knowledge of the emissions is lacking, the a priori emissions considered are negligible and the cost for deviating from them is small.

Each individual HYSPLIT run used in constructing the **TCM** released approximately 20,000 particles representing 1 unit mass over one hour over a cylindrical volume 1 km in the vertical and 1 km diameter centered at the vent. The modeling system does not consider plume dynamics and spreading of the umbrella cloud is only taken into account by utilizing an initial source term above the vent with some width. For this eruption, the initial plume width was not observed to be large, and a relatively small width of only 1 km was employed for most runs.

The HYSPLIT runs covered a vertical area from the vent at 2.88 km to 12.88 km. The HYSPLIT emissions covered a time period from 10/21/2020 19 UTC to 22 October 2020 00 UTC. No emissions outside of this time period or above 12.88 km were considered in rest of the analysis.

A few inversions were performed with heights up to 16.88 km as  Horváth et al. (2021) found that the maximum plume height reached 15-16 km. However results from those runs estimated no significant emissions above 12 km and we do not discuss them further here. This is not surprising as most of the mass is concentrated at the umbrella cloud height rather than the top height and these may differ significantly (Mastin et al., 2009).

Clear sky observations are pixels where no ash was detected. Several options for the inclusion of clear sky observations in the inversion were considered. Figure 3 shows emissions determined with several different time periods and several different ways of utilizing the clear sky values.

When all clear sky observations were included, emissions tended to be lower, especially when using later time periods. The vertical profile was also flatter, with the mass release distributed more evenly throughout the column. Very little mass was emitted at 19:00 UTC or below 4 km or above 12 km.

When no clear sky observations were included, estimated emissions tended to be higher. When using later time periods, the peak in the emissions also tended to occur earlier, around 19:00 UTC which was the earliest time period considered for possible emissions. In the vertical profile there was a strong peak between 7.5 and 10 km but significant mass also emitted below 4 km and above 12 km. Early emissions at 19:00 UTC and significant amounts of mass below 4 km or above 12 km are considered unlikely and the presence of mass at these times and locations indicates that the clear sky observations are important constraints.

A balance was struck when near field clear sky observations a certain distance, such as 2 or 3 pixels, from the observations of ash were excluded. Excluding the near field clear sky observations also made the emissions estimates less dependent on the time periods used in the inversion.

Increasing the width of the initial cloud increased the estimated emissions slightly when some clear sky pixels were utilized as shown in (Figure 3(g) and (h)) When all clear sky pixels were utilized (not shown), emissions were very similar to RunB with all clear sky pixels. The results of the inversion were not particularly sensitive to the eruption column width within this range of 1 to 20 km.

The difference between using 20 and 6 $\mu$m particles is also shown, with basically no difference when using time periods before particle separation begins at 22 October 2020 00 UTC. For the other time periods the determined emissions are quite similar. We attempted to use the inversion to determine a simple particle size distribution by utilizing output from three runs with 6, 20, and 50 $\mu$m sized particles as separate columns in the TCM. The contribution from the 6 $\mu$m size particle was the most dominant and stable, meaning that the emissions amount did not exhibit much dependence on the observational time periods utilized. The other two particle sizes tended to have smaller emission rates and emissions which were quite dependent on the observational time periods utilized.

Figure 4 illustrates the effect of assimilating different time periods in the inversion algorithm. Most of the mass is emitted between 8 and 10 km between 20 and 22 UTC on 21 October. This is consistent with observations. The total amount of mass that is estimated to have been emitted decreases as later time periods are assimilated (Figure 4(e)). Although there could be several reasons for this, we find that this is in large part due to the dispersion of the ash cloud not being adequately represented by the model. The spatial gradient in the observed mass loading is much steeper than that of the model and this disparity becomes greater over time.

For RunM, using the GEFS, we look only at source terms determined using observations up to 00 UTC and some clear sky observations as shown in Figure 5. A separate inversion is performed for each ensemble member. The average emissions are consistent with those estimated in RunB. Two ensemble members, gep09 and gep11, estimated significantly less mass than the others due to less agreement between the simulated and observed ash cloud. In Sect. 5.2 we discuss bias correction of RunM and in Sect. 6 we discuss evaluation of this run. Because only observations up to 0 UTC were utilized in the inversion algorithm, verification of time periods after that can be considered verification of short term forecasts (0 to 12 h).

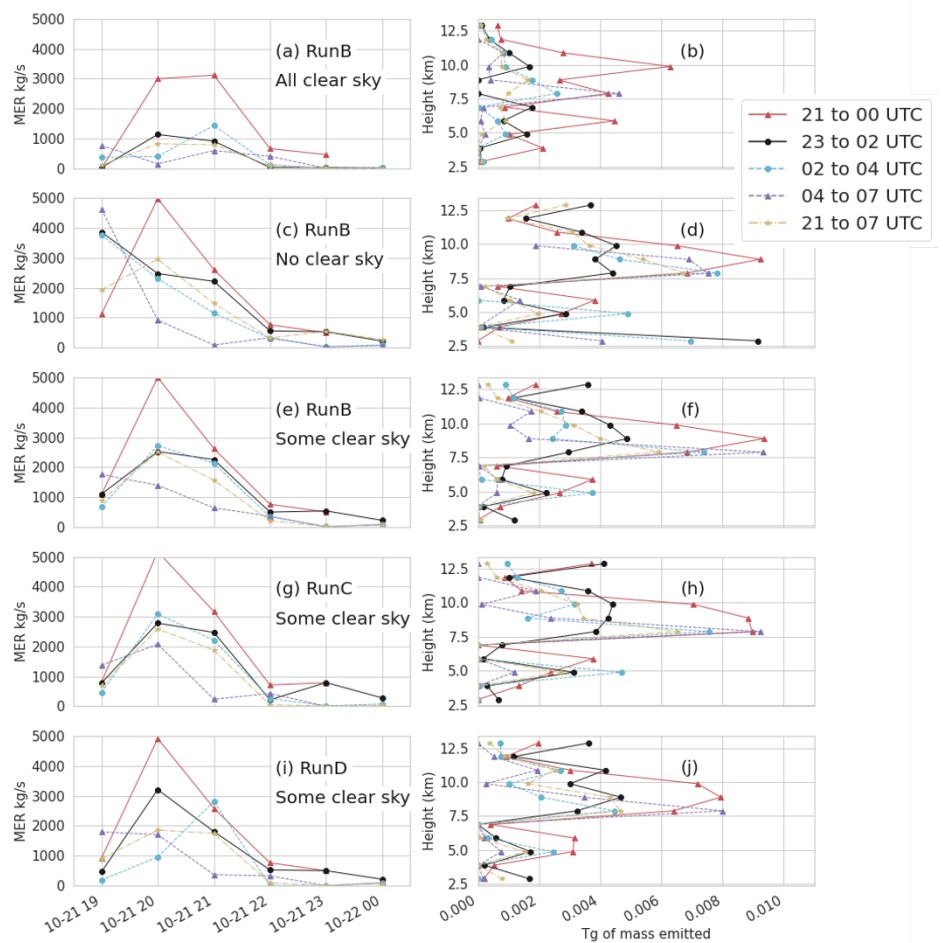

**Figure 3.** Emissions determined from inversion algorithm for several runs as indicated in the text. Left column shows mass eruption rate as a function of time. The time indicates the start time and the $MER_f$ would persist for an hour. Right column shows total mass emitted as a function of height. The height indicates the bottom and the mass would be distributed over 1 km. The legend indicates the hours of observations which were utilized for the inversion. Hours 21 and 23 occurred on 21 October 2020 while hours 00 to 07 occurred on 22 Ocboter 2020. Some clear sky indicates that only clear sky observations farther than 3 pixels from an ash observation were used in the inversion.

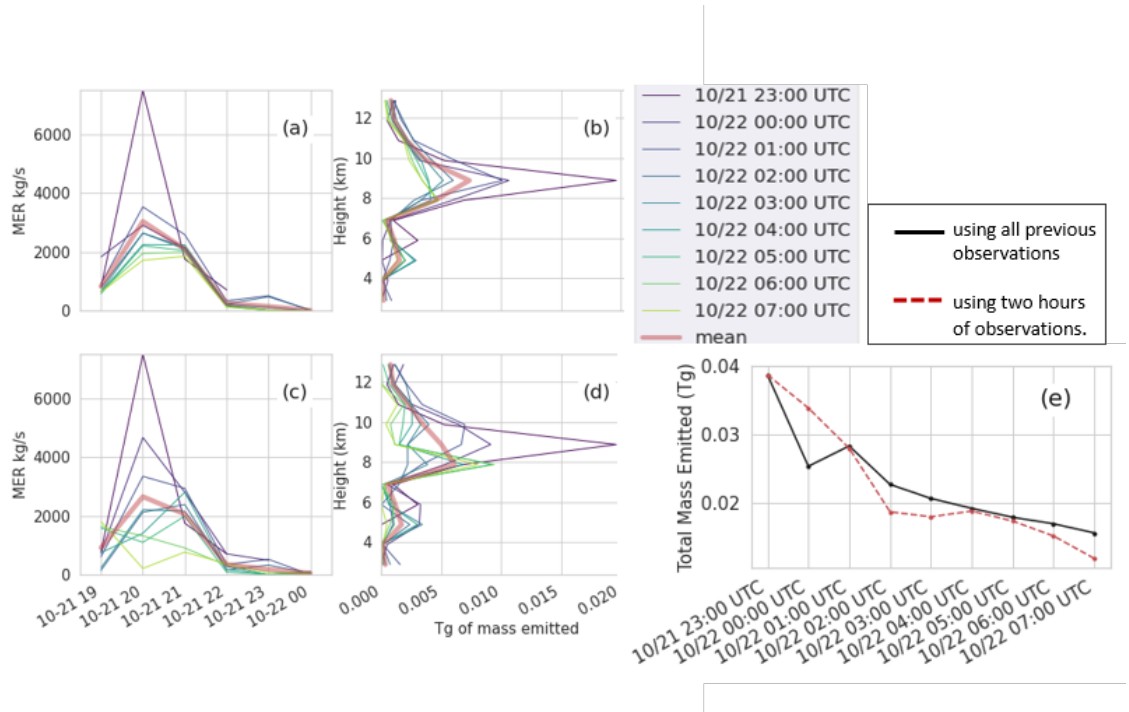

**Figure 4.** Emissions determined from inversion algorithm for RunB. (a,b) are emissions determined using all observations up to the time in the legend. (c,d) are emissions determined using two hours of observations ending at the time in the legend. (a,c) show $MER_f$ as a function of time. (b,d) shows total mass emitted as a function of height. (e) Shows the total mass emitted for each. Clear sky observations farther than 2 pixels from non-clear sky observations were used.

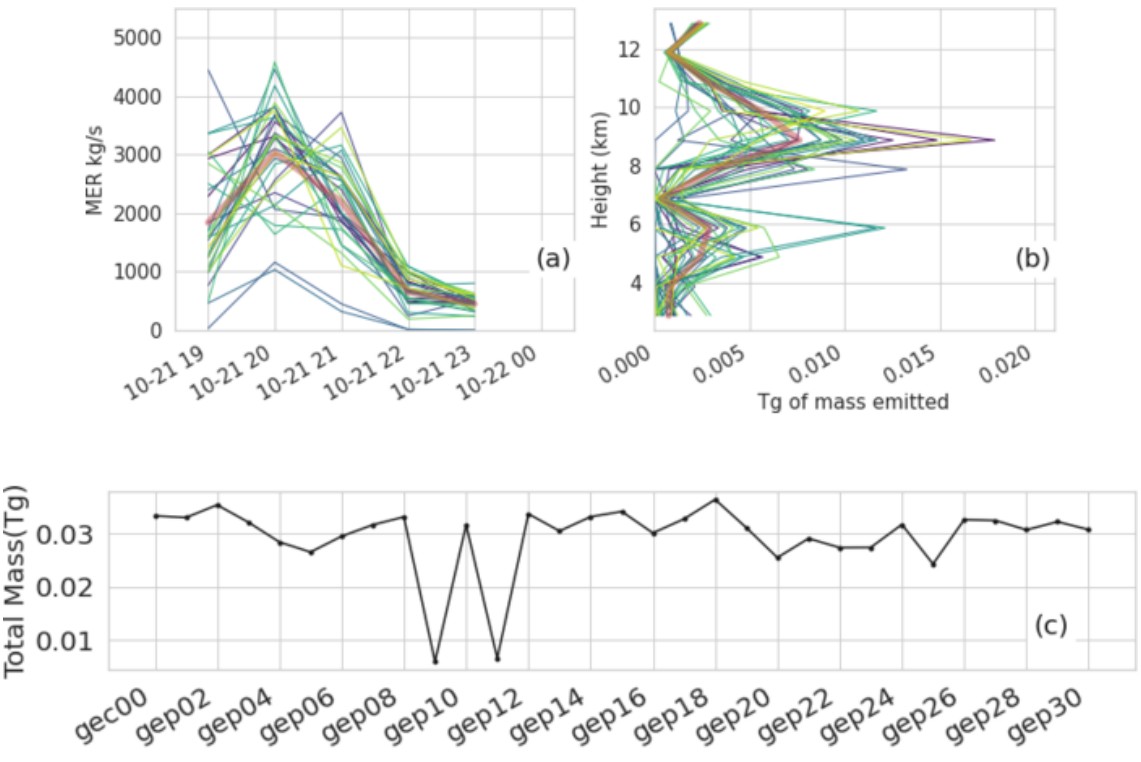

**Figure 5.** Emissions determined from inversion algorithm utilized with all 31 members of the GEFS. (a,b) Emissions determined using all observations up to 22 October 2020 00 UTC. Thin lines in blue, green and yellow show different ensemble members. Thick red line shows average. (c) Shows the total mass emitted for each ensemble member. Clear sky observations farther than 3 pixels from non-clear sky observations were used.

## 5 Bias correction with CDF matching

To correct bias, we employ a procedure called CDF matching (Reichle and Koster, 2004; Piani et al., 2010; Gudmundsson et al., 2012; Belitz and Stackelberg, 2021) which transforms forecast values so that their CDF more closely matches that of the observations. To our knowledge, this is the first time this technique has been utilized for correcting output from an ATDM model. We employ a relatively simple method for creating the transform function. For each time period, the modeled and observed values of mass loading are sorted from greatest to least and then paired so the greatest observed value is paired with the greatest modeled value and so forth. Here pairs in which either the observed or modeled value are 0 are discarded. Usually there are more modeled values above 0 than observed. A linear fit is applied to the difference between the pairs as a function of forecast value. Higher order fits or other functional forms can be used or a non-parametric approach sometimes referred to as quantile mapping can be used (Piani et al., 2010; Gudmundsson et al., 2012). Future work may include comparing different methods of constructing the transform function. We find that a linear fit is adequate and has the advantage that the multiplicative factor determined from the slope has a physical interpretation as a correction to the mass eruption rate. The intercept shifts all mass loading values in the direction opposite the sign of the intercept. The correction is applied at each pixel.

$$s' = (1 - m)s - b \tag{1}$$

Where $s'$ is the corrected mass loading, $s$ is the original mass loading, $m$ is the slope and $b$ is the intercept.

There are several practical considerations in adding or subtracting a constant value to the simulated mass loading or concentration values. Propagating the additive correction to ash concentrations would involve some assumptions such as dividing the correction evenly among the number of vertical levels containing ash.

$$s = \sum_i^n L_i c_i \tag{2}$$

$$s' = (1 - m) \sum_i^n L_i c_i - b \tag{3}$$

$$c_i' = (1 - m) L_i c_i - b_i \tag{4}$$

$$\sum_i^n b_i = b \tag{5}$$

Where n is the number of vertical levels. $L_i$ is the thickness of level $i$. $c_i$ and $c_i'$ are the uncorrected and corrected concentrations at level $i$. $b_i$ is an additive correction applied to the concentration at level $i$ and the sum of $b_i$ over all levels must add to $b$. There are multiple strategies that could be considered for estimating the $b_i$. However, as we are only considering mass loading values in this paper, we will not delve into this further here.

Subtracting a value (positive intercept) is similar to applying a threshold as negative values must be converted to 0. When adding a value (negative intercept), the value is only added to modeled values which have above zero mass loading to begin

with. As these procedures can decrease and increase the spread of the forecast cloud respectively, the intercept can be loosely interpreted as an indicator of how well the spread of the forecast cloud matches that of the observed.

In the next sections we describe the bias correction for RunA and RunM. In section 6 we evaluate how effective CDF matching is at reducing bias. To evaluate whether it can be used to improve the forecast, we apply the correction determined at 00 UTC to later time periods.

## 5.1   Bias correction for RunA

Figure 6(a)–(d) demonstrates the CDF matching procedure for one ensemble member of RunA at two different time periods. Figure 6(e)–(f) shows slope and intercept of the fit for all ensemble members as a function of time. The slope increases slightly over time with a mean value of 0.76 averaged over all ensemble members and times through 12 UTC. The average value from all the ensembles is 0.74 at 0:00 UTC and almost 0.80 at 06:00 UTC. The almost constant value of the slope over time can be 235 translated into a correction to the mass eruption rate. A new $\mathrm{MER}_f$ of $0.25(1.5 \times 10^4 \ \mathrm{kg \, s^{-1}}) = 3.75 \times 10^3 \ \mathrm{kg \, s^{-1}}$ is consistent with the one being returned from the inversion algorithm as described in Sect. 4.3.

The intercept is a fairly large negative number at early times and increases to a small positive number at later times. Negative intercepts shift the CDF to the right by adding a constant. Figure 6(b) illustrates how the multiplicative value of 0.26 shifts the CDF to the left while the additive value of 2.2 shifts it right. The drift from negative to slightly positive values occurs because 240 the forecast cloud is initially more compact than the observations, the high values are concentrated in a smaller area, and then over time it becomes more disperse than the observations.

RunA2 was initialized with a 20 km cloud diameter to see if this would create a more realistic initial condition. The qualitative behavior of the slope and intercept of the fits was the same as shown in Figure 6. The slope was generally a bit lower than RunA with an average value of 0.68 over all ensemble members and time periods and 0.65 at 0:00 UTC and increasing to about 245 0.74 at 06:00 UTC. Intercept values at the early times, before 0:00 UTC did become less negative, however intercept values at later times became more positive. Increasing the spread of the cloud at early times to more closely match observations, caused the spread of the cloud at later times to have a larger mismatch.

The modeled horizontal dispersion of the cloud is too fast. As discussed in Sect. 4.3, this is also indicated by the behavior of the inversion algorithm when utilizing different time periods of observations. In addition,  de Leeuw et al. (2021) and Cai et al. 250 (2021) found a similar issue in forecasting $SO_2$ emissions from Raikoke with the Numerical Atmospheric-dispersion Modeling Environment, NAME, and Massive-Parallel Trajectory Calculations, MPTRAC, models respectively.

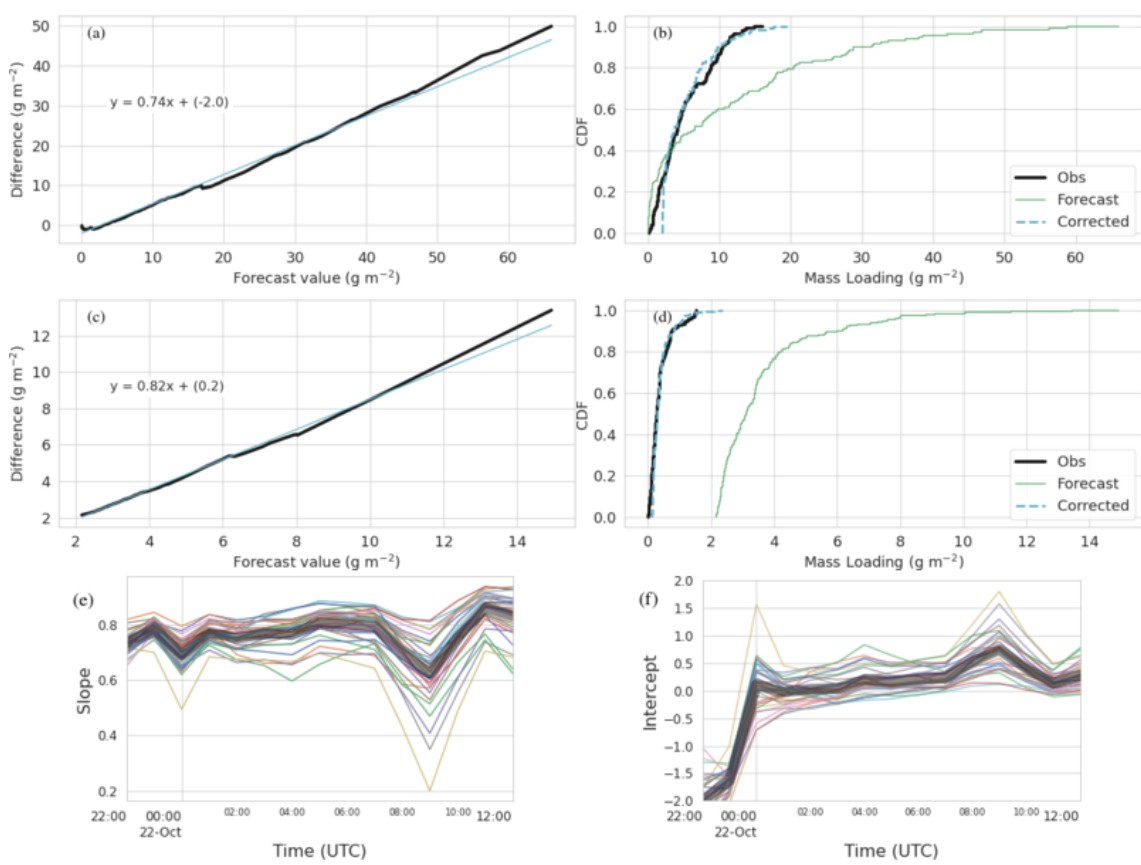

**Figure 6.** Demonstration of CDF matching for RunA. (a,b) CDF matching for time period 21 October 2020 22–23 UTC and ensemble member gec00. (c,d) CDF matching for time period 22 October 2020 10–11 UTC and ensemble member gec00. (a,c) show the linear fit (cyan) to the difference between observed and forecast values (thick black line). (b,d) shows the CDF of the forecast values, observed values, and corrected forecast values. (e) shows the slope and (f) shows the intercept of the linear fit as a function of time for each ensemble member. The thick gray line is the mean of all the ensemble members.

## 5.2 Bias correction for RunM

As discussed in more detail later, even the ensemble produced using the source terms derived from the inversion algorithm, RunM, was biased. We investigated whether utilizing the CDF matching technique could effectively reduce the bias for this case.

Figure 7 shows slopes and intercepts from the fits to RunM. As the initial bias was much smaller than for RunA, the slopes are much smaller as well, with the average slope around 0 through about 7 UTC. There are also some negative slopes, which indicate an increase in mass eruption rate. The gep09 and gep11 ensemble members shown in cyan and orange in Figure 7 have more negative slopes and intercepts than most of the other ensemble members. This is consistent with the inversion algorithm estimating a lower mass eruption rate for them which the CDF matching then attempts to correct.

The trend for the intercepts is very similar to that seen for RunA. Values start negative for the times which have been utilized in the inversion algorithm (before 0 UTC) and then become positive for most of the ensemble members. This indicates that even with the improved source term, the modeled cloud is initially more compact than observed and then becomes more dispersed than observed. In later sections we will see that the positive values of the intercept at later times are indicative of a high bias in the ensemble.

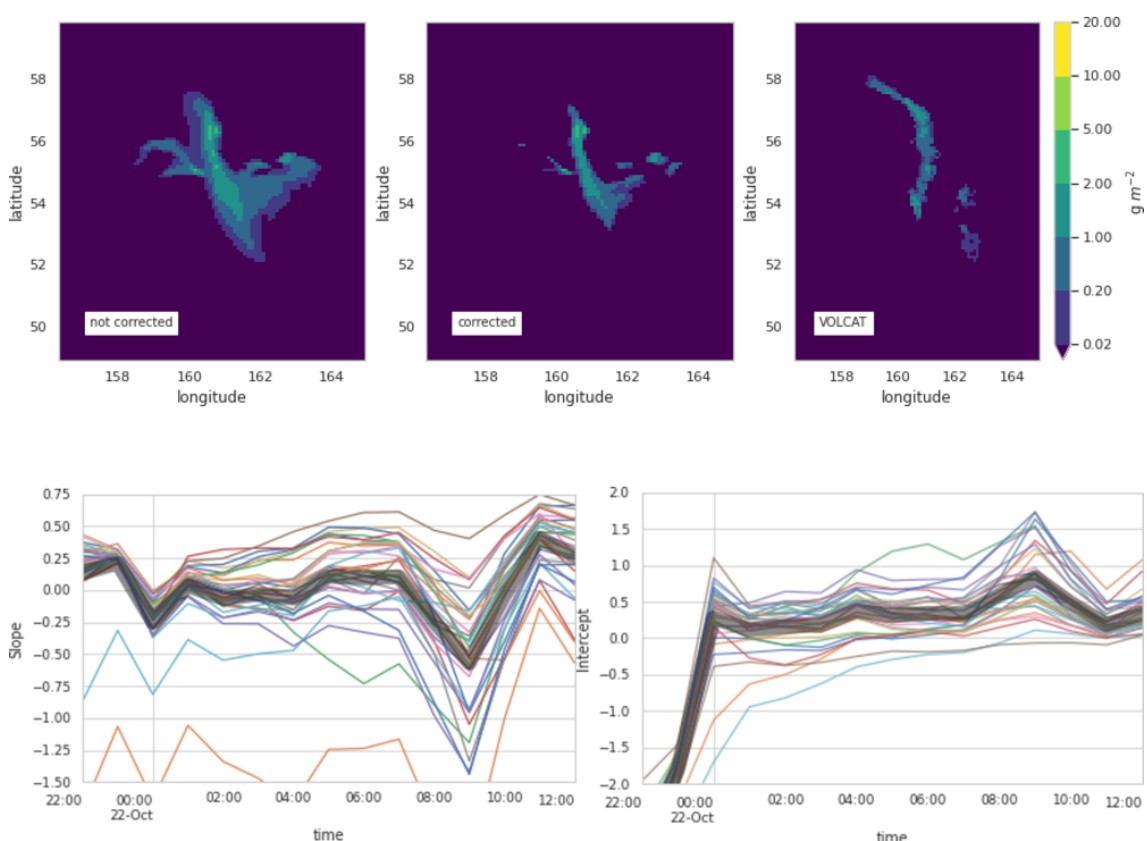

**Figure 7.** Demonstration of CDF matching for RunM. The top row shows column mass loading for VOLCAT (right) and ensemble member gep12 on 22 October 2020 06–07 UTC with bias correction (middle) and without (left). The bottom left shows the slope and bottom right shows the intercept of the linear fit as a function of time for each ensemble member (different colored lines). The thick gray line shows the mean.

## 6 Evaluation

We make the assumption that verification of modeled column mass loading values can be used as a proxy for verification of forecast concentrations. The reason for this is practical as column mass loading values are now generally widely available for many eruptions of this size and larger while measurements of concentrations are not. The validity of this assumption may be investigated in the future by employing data from lidar which can give information on ash cloud thicknesses, or by utilizing any in-situ measurements that may be available from aircraft flights.

Table 2 summarizes the measures which will be discussed in detail in the following sections.

**Table 2.** Summary of measures categorized by the main aspect of forecast performance they evaluate.

| Name | Summary |
|---|---|
| **Are the forecast probabilities well calibrated?** | |
| Reliability Diagram | made up of the refinement distribution and calibration function |
| Refinement distribution | histogram which indicates forecast sharpness |
| Calibration Function | ideally lies along 1:1 line |
| **Does the ensemble satisfy the consistency condition?** | |
| Rank histogram | flat histogram is ideal |
| **Is the forecast better than a reference forecast?** | |
| Brier Skill Score (BSS) | 0 (worst) to 1 (best) |
| **At what spatial resolution does the forecast have skill?** | |
| Fractions Skill Score (FSS) | Similar to BSS as a function of spatial resolution |
| | 0 (worst) to 1 (best) |
| **What is the frequency bias of the forecast?** | |
| Asymptotic Fractions Skill Score (AFSS) | 0 (worst) to 1 (best) |
| **What probability threshold is appropriate for risk-based approaches?** | |
| **How well does the forecast classify above and below mass loading threshold areas?** | |
| Precision recall curve (PRC) | plots precision vs POD for different probability thresholds |
| precision | 0 (worst) to 1 (best) |
| probability of detection (POD) | 0 (worst) to 1 (best) |
| Receiver operating characteristic (ROC) | plots POD vs POFD for different probability thresholds |
| probability of false detection (POFD) | 0 (best) to 1 (worst) |
| **General measure** | |
| Area under the PRC curve (AUPRC) | 0 (worst) to 1 (best) |
| **Method to spatially coarsen the probabilistic forecast** | |
| Neighborhood ensemble probability (NEP) | used in FSS, AFSS, and PRC |

## 6.1 Setting thresholds

The concentration thresholds of interest for aviation are 0.2, 2, 5 and 10 mg m$^{-3}$ (Prata et al., 2019). Proxy mass loading
thresholds are formed by assuming that in some cases ash layers are approximately 1 km thick and then the thresholds have
the same values but with units of g m$^{-2}$. In some cases, it may be desirable to set the proxy mass loading thresholds according
to the vertical resolution of the model output. For instance, with a vertical resolution of FL50 (flight level 50), approximately
1.5 km, modeled average concentrations of 0.2 mgm$^{-3}$ which are one layer thick would result in mass loadings of 0.3 g m$^{-2}$.

## 6.2 Qualitative comparison

A qualitative comparison between modeled and forecast data provides context for discussion of the verification metrics. Consequently we start the discussion with a side by side comparison of composited VOLCAT observations and HYSPLIT model output of ash column mass loadings shown in Figure 8. Six time periods are shown which range from 4 hours to 16 hours after the eruption start time. The evolution of the ash cloud can be described as follows.

Part of the ash cloud is stretched in the north-south direction. This piece initially is located to the west of the volcano, but gradually moves to the east. As it moves east, it becomes longer and thinner and forms a bow, as the parts to the north and south of the volcano move more slowly. By 12 UTC, this line of ash has broken into three small areas, one just to the east of the volcano which has the highest retrieved top heights of around 10-11 km, one to the south with the lowest retrieved top heights of between 5-7 km, and one to the northwest with retrieved top heights of between 7-9 km. Presumably there might be ash at very low concentrations still connecting the pieces.

At 00:00 UTC, another part of the cloud is located to the southeast of the volcano. This portion breaks off from the piece discussed above by 04:00 UTC and moves to the southeast. By 08:00 UTC, this piece is no longer observed.

The model run shown reproduces the general trend fairly well. However, the placement and shape of the line of ash is not reproduced perfectly. In many of the model runs, including this one, the simulation does not stretch the line far enough north. Nor is the placement and shape of the southeast piece of ash correct. The two pieces of ash remain attached in the model runs for a longer period of time which results in a v-shape for the modeled cloud. The southeast piece of ash remains above threshold for much longer in the simulation.

In general, all the individual model runs follow these trends. Model runs which utilize the inversion algorithm for source determination tend to show better qualitative agreement. However, the exact evolution of the ash cloud is not faithfully reproduced by any run.

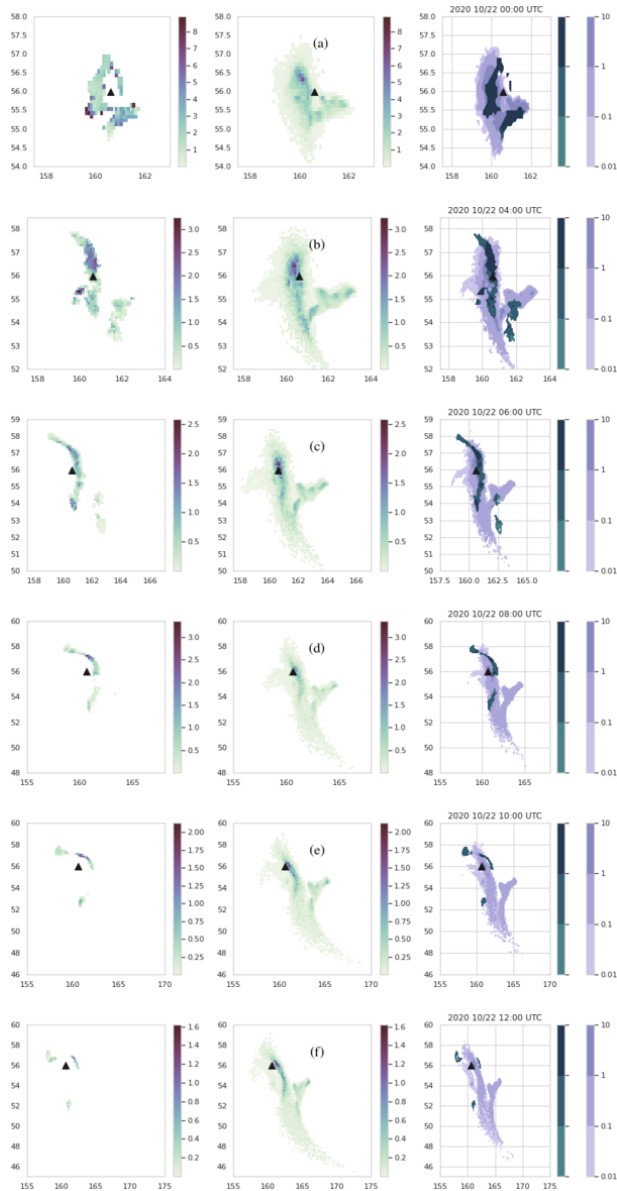

**Figure 8.** Evolution of observations and model results for RunB for emissions determined from inversion algorithm incorporating three hours of observations from 21 October 21 UTC through 22 October 00 UTC. The time stamp for each indicates the beginning of a one hour average. Thus the top row is a 1 h forecast, the second row is a 4 h forecast and so forth. The black triangle shows the location of the vent. Note that the color scale changes for each time period. Left panel shows composited VOLCAT data. Middle panel shows model output. Left and middle panel have same color scale. Right panel shows composited VOLCAT data in dark green while the modeled data is in light purple. Units on all colobars is g m$^{-2}$.

## 6.3 Agreement in threshold level and agreement in percentile level

For probabilistic forecasts we follow Galmarini et al. (2004). Agreement in threshold level (ATL) is the normalized number of ensemble members which predict values above a given threshold. It can also be referred to as ensemble relative frequency of exceedance. Figure 9(c) shows number of model runs exceeding a given mass loading threshold. Figure 9(e) shows an example of the agreement in percentile level (APL) which is the value at each location for which a given percentage of the model runs are smaller. This example shows concentrations for which 84 % of the runs (26/31) gave smaller concentrations and 16 % of runs (5/31) gave larger concentrations. An APL with percentile of 50 % would provide the median values and APL with percentile of 100 % would provide maximum values.

A binary predictand can be recovered from the ATL by applying a probability threshold. In Figure 9(d) a probability threshold of 16 % (5/31 ensemble members) is applied to (c). A binary predictand can be recovered from the APL by applying a mass loading (or concentration) threshold. In Figure 9(f) a threshold of 0.2 g m$^{-2}$ was applied to (d). As shown, applying a probability threshold of 100-APL to the ATL for a given mass loading threshold (Figure 9(d)), is equivalent to applying the quantitative threshold to the APL (Figure 9(f)).

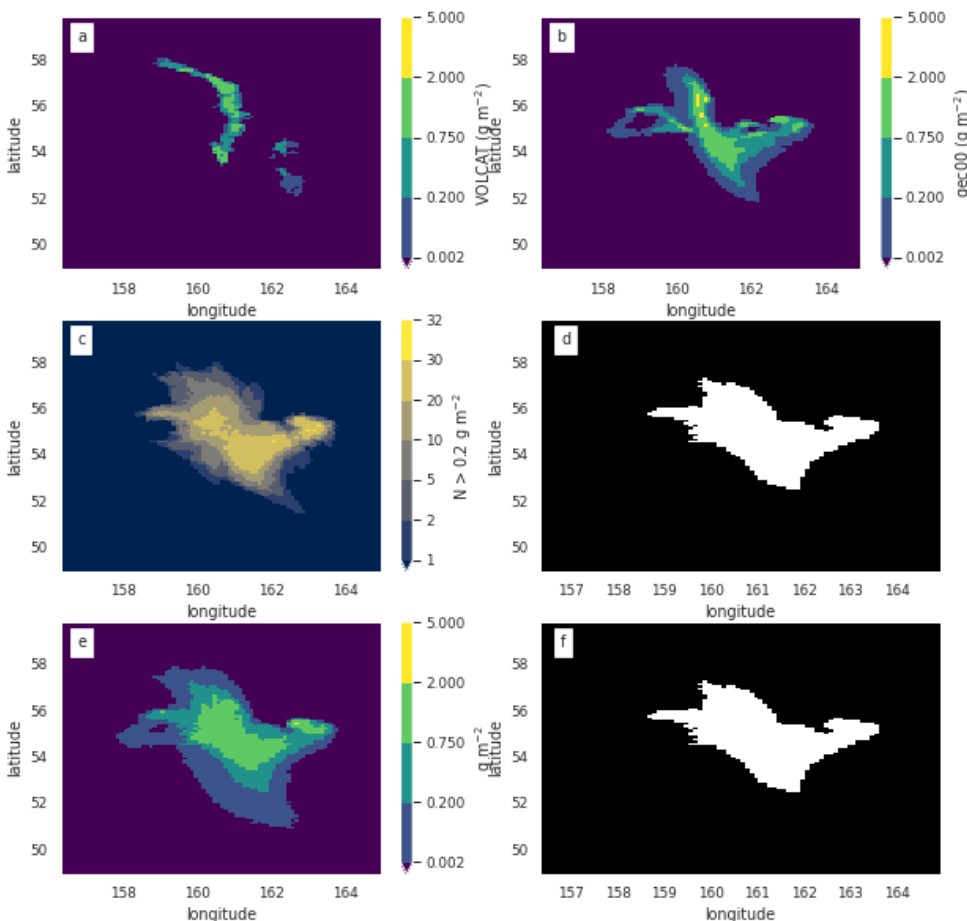

**Figure 9.** Observations and model forecast for 22 October 2020 06 UTC. (a) VOLCAT observations. (b) One member of the HYSPLIT dispersion ensemble using the gec00 control run from the GEFS meteorology. (c) Number of ensemble members showing mass loading above $0.2 \ \mathrm{g \ m^{-2}}$. If normalized by the number of ensemble members (31), this would be the ATL. (d) Areas above a 16 % ensemble relative frequency threshold (> 5 members) (e) Concentrations at APL of 84 %. Note that the color scale is designed to convey information relevant to the thresholds described in Section 6.1 (f) Areas above $0.2 \ \mathrm{g \ m^{-2}}$ threshold in the APL of 84 %. No bias correction applied to the simulations.

## 6.4 Dosage and ash cloud extent

Some sources indicate that it is dosage rather than concentrations that are the relevant factor for airlines (Prata et al., 2019; Hirtl et al., 2020). To compute a dosage from gridded concentrations just sum over the time spent in each grid cell, $t_i$, multiplied by the concentration in the grid cell, $C_i(t)$.

$$D = \sum_i t_i C_i(t_i) \tag{6}$$

If velocity is constant then D is not sensitive to spatial averaging that is performed parallel to the flight path.

For probability of exceeding a critical dosage $D_c$, this equation should be applied to each ensemble member and ensemble relative frequency of exceeding the dosage computed from the resulting ensemble of dosages, $P(D > D_c)$.

The most straightforward way to verify forecast $P(D > D_c)$ would be to use data from aircraft fitted with sensors that can measure ash concentrations. However, until such data become widely available, we might ask what the verification of probability of exceedances of mass loadings can tell us about probability of exceedances of dosages along a flight route.

The probability of exceeding a dosage through a certain grid cell, $x_j$, can be related to the probability of exceeding a concentration in that grid cell. However, combining these utilizing the probability of exceeding a concentration or ATL to get probability of exceedance of dosage along a route containing multiple grid cells is not possible because information about the relationship between concentrations in adjacent grid cells for each member is not preserved by the ATL field.

A dosage could also be computed from an applied percentile level as in Hirtl et al. (2020) which utilizes the 75 % percentile level to produce a concentration field similar to that shown in Figure 9(e). This seems straightforward, but as for the APL, the relationship between concentrations in adjacent grid cells for each member is not preserved by the APL field and it isn't clear what such a dosage represents. If the ensemble members have significant spread in space, as occurs here, the dosage would be too conservative for a high APL as the planned route would essentially take the plane through multiple instances of the cloud. On the other hand if too small an APL is used, high concentrations in an area may be missed entirely. As each ensemble member may predict a high concentration but in a slightly different place, the probability that a high concentration somewhere in the area exists could be high, while the probability at each spatial location is low.

If dosage is the relevant quantity, then predicting the extent of the ash cloud accurately is critical. Assuming airspeed is around $v = 250$ m s$^{-1}$ and dosage threshold is around $D_t = 14.4$ g s m$^{-3}$ (Prata et al., 2019) then distance the aircraft can safely travel through ash is given by

$$d = \frac{D_t}{C} v \tag{7}$$

If this is the case, even if the largest concentrations that are expected in the distal ash cloud are 10 mg m$^{-3}$ then only ash areas with a width larger than d=360 km, about 4 degrees, are of concern.

Under these circumstances, accurately predicting the spatial location of small areas of ash cloud may not be particularly important. Instead predicting the time at which the ash cloud is small enough to no longer be of concern becomes important.

Simply comparing the area of the observations to area of the simulations over time as in Figure 10 provides information on

if the simulation is capturing the evolution and end of life of the ash cloud correctly. Without bias correction, both RunA and RunM overpredict the lifetime of the ash cloud significantly. The extent of lower mass loading of ash, $0.2$ g m$^{-2}$ continues to increase through 12 UTC (Figure 10(a) left column). In contrast, the extent of the higher mass loadings follows the observations more closely (Figure 10(a) right column). This mismatch occurs because the spatial gradient of the mass loading field is much steeper in the observation than in the simulation as can be seen in Figure 8. This could be an indication that the modeled

turbulent dispersion which controls the spread of the ash is not reproducing what is observed.

Applying the bias correction brings the simulated areas of both the lower and higher mass loadings in line with the observed area (Figure 10(b)). For some ensemble members the bias correction produces much too high areas at some times. This is because a positive shift sometimes increases the area above the threshold too much.

Applying the bias correction calculated at 00 UTC to the forecasts also improves the estimation of the total area above

threshold significantly for many of the ensemble members. For some members in which the bias correction has a positive shift at 00 UTC, the forecast area does become too large. This can be ameliorated by applying only the multiplicative correction when the shift is positive.

We will see later that allowing the positive shift helps ensure that a high probability of detection can be achieved which is important for a strategy of total avoidance of areas of ash.

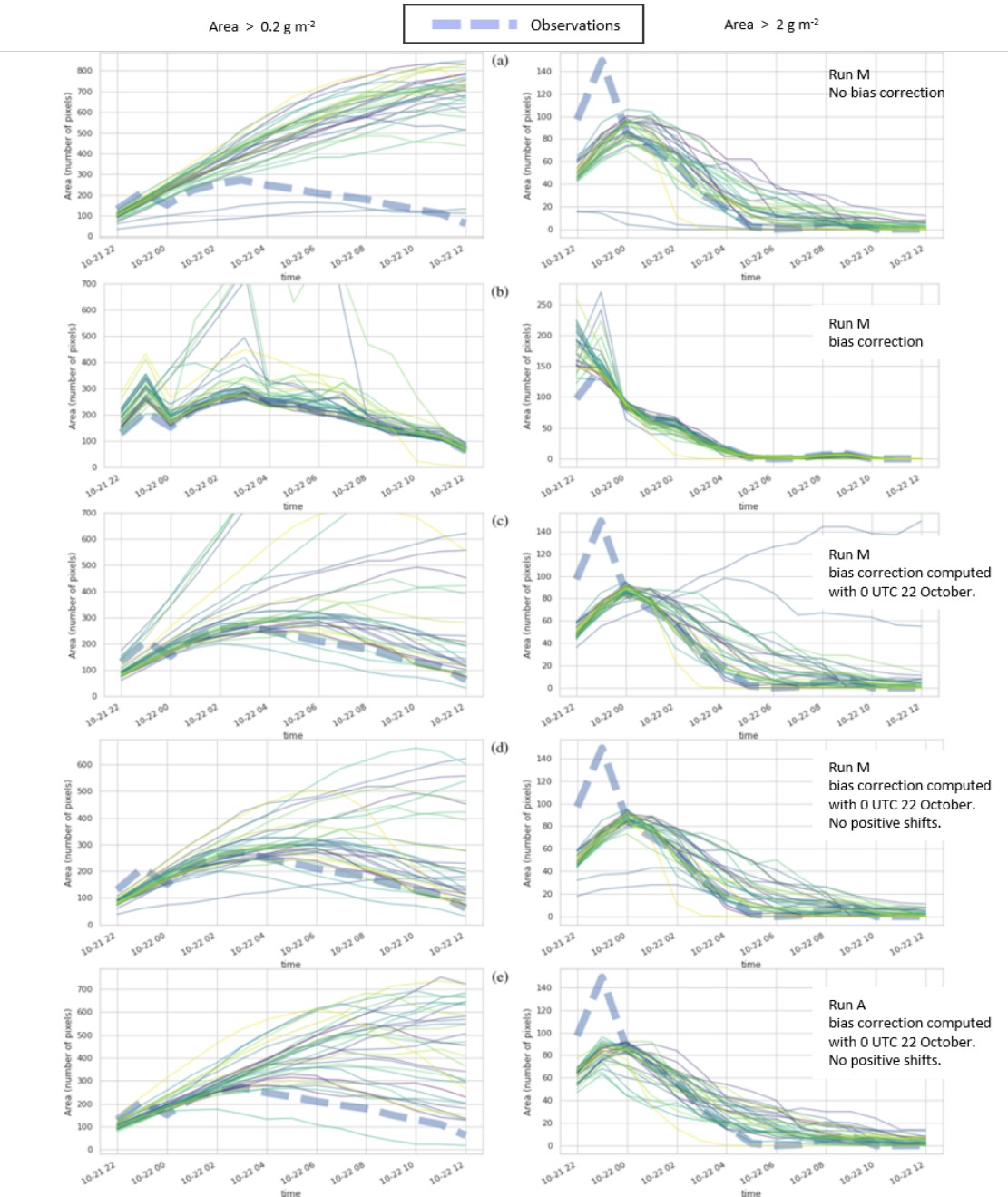

**Figure 10.** Area above threshold as a function of time. Left column shows area above 0.2 g m$^{-2}$ threshold and right column shows area above 2 g m$^{-2}$ threshold. Pixel size is $0.1 \times 0.1^o$. Thin lines are each ensemble member. (a) RunM with no bias correction. (b) RunM with bias correction computed at each time period. (c) RunM with bias correction computed at 22 October 00 UTC. (d) RunM with bias correction computed at 22 October 00 UTC and no positive shifts allowed. (e) RunA with bias correction computed at 22 October 00 UTC and no positive shifts allowed.

## 6.5 Rank histogram and reliability diagram

The reliability diagram, which is made of up the refinement distribution and calibration function, illustrates how well calibrated the modeled probabilities are. The diagram is computed from the ATL field which shows the modeled probability of exceeding a given threshold at each observation point. The modeled probabilities take on values $y_i = \frac{i}{N_{\text{ens}}}$, where $i$ is an integer between 0 and the number of ensemble members, $N_{\text{ens}}$, which is 31 here.

The refinement distribution is a histogram of how often each modeled probability, $y_i$, occurred. It provides information on the sharpness of the forecast. According to Wilks (2011a) "Forecasts that rarely deviate much from the climatological value of the predictand exhibit low sharpness....By contrast, forecasts that are frequently much different than the climatological value of the predictand are sharp." As the presence of volcanic ash in the atmosphere is relatively rare, we take the climatological value to be a low probability, near 0 %. Thus we will describe forecasts with many values of large $y_i$ as sharper than those with few values of large $y_i$.

Refinement distributions for various time periods and mass loading threshold levels are shown in the right hand column of Figures 11 and 12. These figures are the same except Figure 11 is for RunM while Figure 12 is for RunA. We only look at 0.1, 0.2, and 2 g m$^{-2}$ thresholds because there are few points for higher threshold levels. The number of counts at $y_i = 0$ is the number of times no ensemble member predicted an above threshold value. This number is domain dependent, but the number of counts for the rest of the bins is not. For the earlier time periods, overlap between the ensemble members was common and the ensemble often expressed high confidence (near 100 %) that values were above threshold. For later forecast times, the frequency with which the ensemble has high confidence in the forecast decreases. For example in Figure 11(b) there is never more than 60 % agreement among ensemble members that mass loading is above 2 g m$^{-2}$. These later forecasts can be described as less sharp. In general, bias correction decreased the sharpness of the forecasts. This occurred because the reduction in the area above threshold also reduced overlap between ensemble members.

The calibration function is $p(o_1|y_i)$, giving the probability of observing the event defined as $o_1$ given the modeled probability of the event, $y_i$ (Wilks, 2011a). Here the event, $o_1$, is that the mass loading exceeded the given threshold. Calibration functions for different mass loading thresholds are shown in the middle columns of Figures 11 and 12. To calculate each point on the calibration function, the fraction of times the observation was above the given threshold is calculated for each $y_i$. If the number of $y_i$ points is low, then the estimated value of $p(o_1|y_i)$ will have more uncertainty. Ideally, the calibration function lies along the 1:1 line. For example, we would expect that if we look at all the times the modeled probability was 50 %, half the time the event would be observed and half the time it would not be observed. When the function lies below the 1:1 line, as it does here, the modeled probabilities are overconfident. For instance a point at (0.80,0.50) means that when the model predicted there was an 80 % chance of occurrence, the actual event was observed only 50 % of the time. The bias correction did not have much of an effect on the calibration function. The high confidence forecasts remained overconfident. The improvement shows up only in the refinement distribution, where we saw that the bias correction reduced the number of these high confidence forecasts or the sharpness. Sharp forecasts are only desirable if they are reliable, so the reduction in sharpness was appropriate.

The rank histogram evaluates whether the ensemble satisfies the consistency condition (Wilks, 2011a, b). The rank histograms shown in the left columns of Figure 11 and 12 are constructed from all points at which either the observation or at least one of the simulations was above $0.1$ g m$^{-2}$ and are constructed from forecasts for multiple time periods. The values at each grid point of each simulation as well as the observation are ordered from least to greatest and the rank of the observation is recorded in a histogram. When the observation has the same value as several simulations, then the count for that observation is divided evenly among those bins. This occurs frequently in this case because there are many points at which the observation as well as multiple simulations have 0 mass loadings.

Without bias correction, the rank histograms for RunM and RunA show a high over-forecasting bias at all time periods. Too frequently the observation is the lowest or one of the lowest values, which is often 0. The bias correction procedure reduces the over-forecasting bias significantly. The use of the inversion algorithm for source determination also improves the rank histogram.

With the bias correction in place, the rank histogram exhibits a U-shape at the earliest times. The U-shape indicates that the ensemble is overconfident. The ensemble members overlap with each other more than they overlap with the observations. This shows up in the reliability diagram as a line which tends to be flatter than the 1:1 line so that high modeled probabilities correspond to lower actual probabilities.

At later forecast times, the rank histogram with bias correction becomes flatter towards the left of the distribution and then tends to smoothly decrease toward the right half of the distribution, with an uptick toward the second to last bin and an abrupt decrease in the last bin. The flatness on the left hand side is mainly due to areas in which the observation and many ensemble members show below threshold mass loadings. The smooth decrease with the slight uptick toward the right is due to the ensemble members overlapping more with each other than with the observation. The uptick occurs in the second to last bin rather than the last bin because a few ensemble members had a high frequency bias at later times due to the addition of a constant in the bias correction. Thus the observation was rarely the highest value, but often the second highest value. If the bias correction was performed so as not to allow addition of a constant, then the uptick occurred in the last bin.

As forecast time increases, the ensemble members overlap less and less with each other which is indicated in the refinement distribution. The calibration function becomes flat with all simulated probabilities corresponding to a low actual probability. The rank histogram becomes quite flat on the lower end indicating a large number of points with below threshold value for the observations and more than half the ensemble members. This is due to increasing difficulty in predicting the location of the ash.

Utilizing the inversion algorithm to determine the source term improved the rank histogram and reliability diagram.

As time passes, the forecast approaches but does not reach a situation in which none of the ensemble members overlap with eachother or with the observation. In such a situation, the rank histogram will be perfectly flat if the average area covered by the ensemble members is the same as the area covered by the observations. If on average, the ensemble members covered more area, but there was still no overlap, then the first bin in the rank histogram would be populated more indicating over-forecasting bias, while if the ensemble members covered less area on average then the last bin in the rank histogram would be populated more indicating under-forecasting bias.

The rank histogram might look very good in such a situation, indicating that the ensemble is providing accurate information on the size of the ash cloud. However the reliability diagram would reveal that the ensemble is not able to provide information on the actual location of the ash cloud. The refinement distribution would show that the forecast has no sharpness. Values of $y_i$ would be limited to only 0 and $\frac{1}{N_{\text{ens}}}$ %. The calibration function would consist of only two points. The point at $y_i = 0$ would be associated with a finite chance of an observation while $p(o_1|\frac{1}{N_{\text{ens}}}) = 0$.

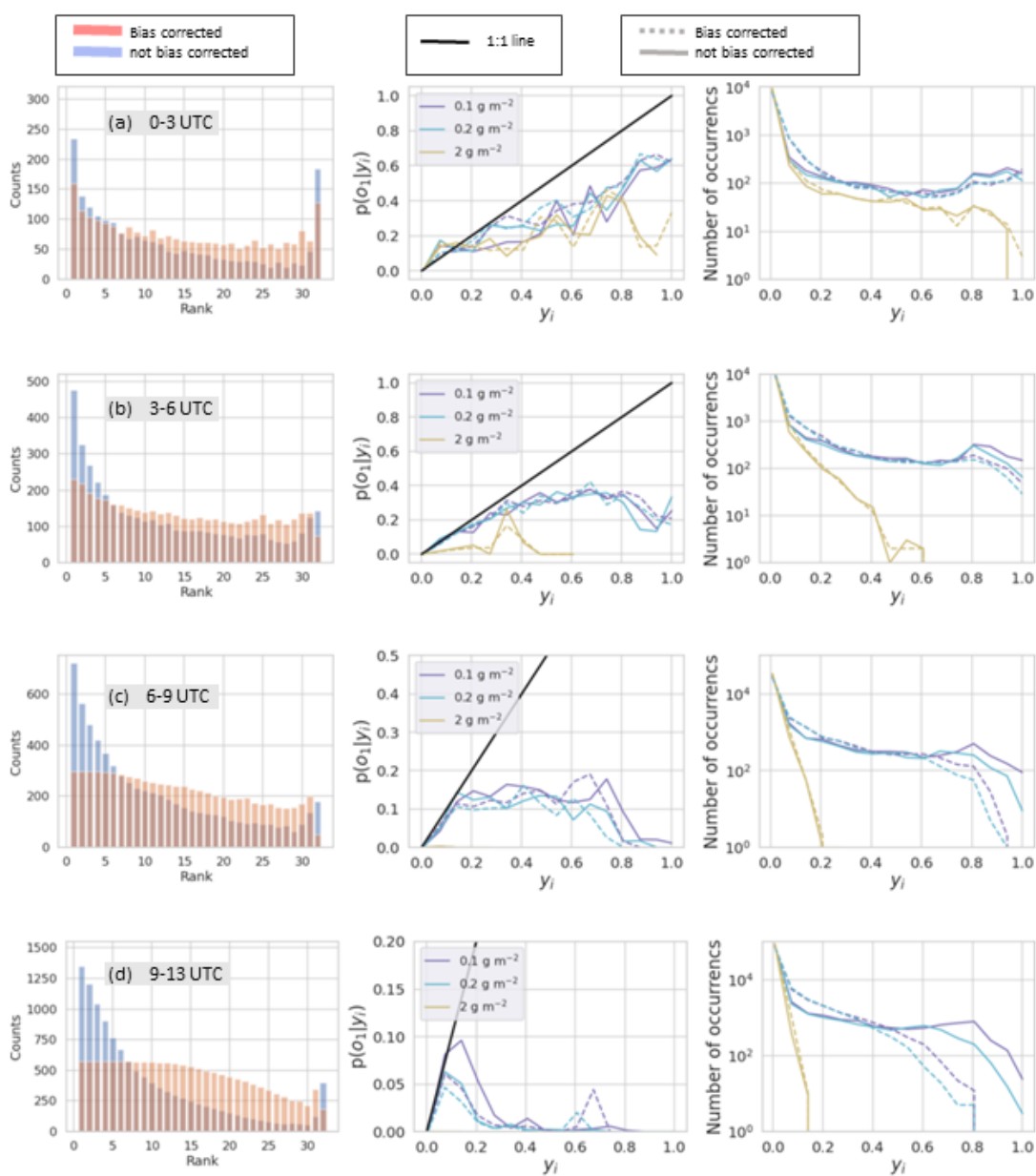

**Figure 11.** Rank histograms (left column), calibration function (middle column) and refinement distribution (right column) for RunM without bias correction (blue bars and solid lines) and with bias correction (red bars and dotted lines). Bias correction determined at 00 UTC and allowing positive shifts. Rank histograms are for all points above 0.1 g m$^{-2}$. Calibration function and refinement distributions are for three different thresholds shown in legend. Each row is for a different time period on 22 October 2020 as indicated in the gray box.

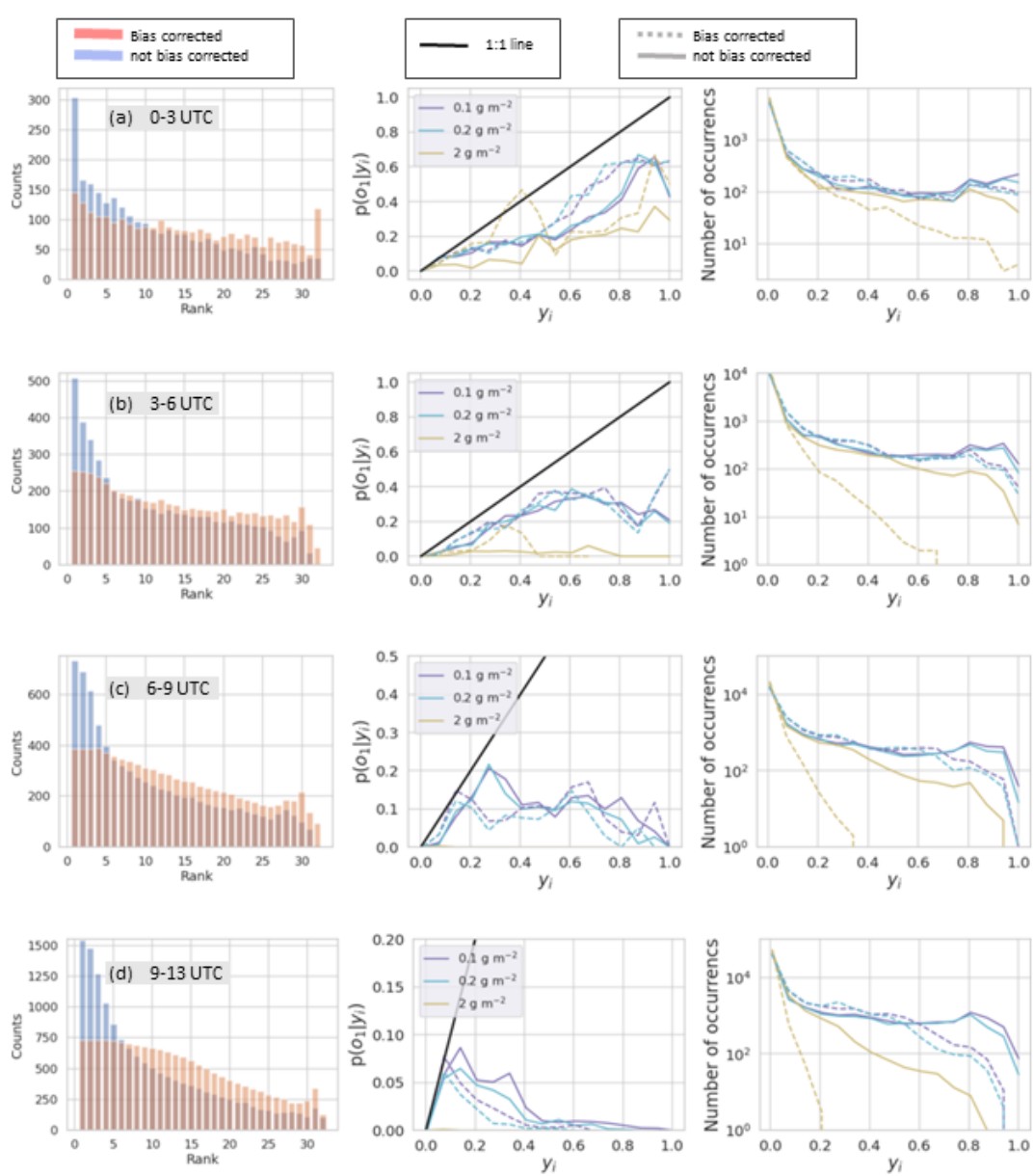

**Figure 12.** Rank histograms (left column), calibration function (middle column) and refinement distribution (right column) for RunA without bias correction (blue bars and solid lines) and with bias correction (red bars and dotted lines). Bias correction determined at 00 UTC and allowing positive shifts. Rank histograms are for all points above 0.1 g m$^{-2}$. Calibration function and refinement distributions are for three different thresholds shown in legend. Each row is for a different time period on 22 October 2020 as indicated in the gray box.

## 6.6 Fractions Skill Score and Brier Skill Score

So far we have seen that, especially with bias correction, the model ensemble can capture the area and qualitative structure of
the observed ash cloud, but struggles to capture the exact placement and shape. By utilizing the fractions skill score (FSS) we
can investigate if the forecasts become more skillful at a different spatial scale.

FSS determines the resolution at which the forecast has skill. It was developed to evaluate precipitation forecasts (Roberts
and Lean, 2008), but has also been used to evaluate volcanic ash forecasts (Harvey and Dacre, 2015; Dacre et al., 2016).
For deterministic forecasts, the gridded data with domain size of $N_x \times N_y$ are converted to a binary field using a threshold.
Observed and modeled fractions are then computed for different neighborhood sizes, n. Here that is achieved by convolution
of the field with a square kernel of size n. The result can be interpreted as gridded data of the same size and resolution in which
the fraction indicates the probability of finding ash above threshold within an area $n \times n$ centered at that pixel. The mean square
error, MSE, skill score is then computed for each n and the FSS is computed by comparing the MSE of the forecast with a
reference forecast.

$$\text{MSE}(n) = \frac{1}{N_x N_y} \sum_{i=1}^{N_x} \sum_{j=1}^{N_y} \left( (O(n)_{i,j} - m(n)_{i,j} \right)^2 \tag{8}$$

$$\text{FSS}(n) = 1 - \frac{\text{MSE}(n)}{\text{MSE}(n)_{\text{ref}}} \tag{9}$$

Where $O(n)$ is the modeled fraction, $m(n)$ is the modeled fraction, $N_x$ and $N_y$ are the number of grid cells in the $x$ and $y$
directions and $N$ is used hereafter to refer to the larger of these.

The MSE described in Roberts and Lean (2008) is the same as the Brier Score (BS) and the FSS is equivalent to the Brier
Skill Score (BSS) as defined in Wilks (2010, 2011a) and utilized by Zidikheri et al. (2018); Zidikheri and Lucas (2020). The
BSS is commonly used to evaluate probabilistic forecasts.

When computing the FSS, it is standard for the reference forecast to be defined as the largest possible MSE that can be
obtained from the forecast and observed fractions (Roberts and Lean, 2008).

$$\text{MSE}(n)_{\text{ref}} = \frac{1}{N_x N_y} \sum_{i=1}^{N_x} \sum_{j=1}^{N_y} \left( O(n)_{i,j}^2 + m(n)_{i,j}^2 \right) \tag{10}$$

Although the FSS generally has been used with a deterministic forecast as a starting point, there is no reason that the modeled
field, $m_{i,j}$, cannot be probabilistic output consisting of values between 0 and 1. This combines the probability of the ensemble
predicting an above threshold event at the grid square with the probability of the model predicting an above threshold event
within a certain area. Ma et al. (2018) refers to this as the neighborhood ensemble probability (NEP) and found that it generally
performed better for precipitation forecasts than the ensemble mean.

At some value of n <= 2N-1, the forecast becomes uniform; the same fraction is in each grid square and these fractions are proportional to the fraction of observed values, $f_o$, and the fraction of modeled values, $f_m$. At this point the value of the FSS is given by the asymptotic fractions skill score (AFSS).

$$\text{AFSS} = 1 - \frac{(f_o - f_m)^2}{f_o^2 + f_m^2} = \frac{2 f_o f_m}{f_o^2 + f_m^2} \qquad (11)$$

If there is no frequency bias, then AFSS=1. However, unless a pixel matching technique is used (Harvey and Dacre, 2015;
Dacre et al., 2016), generally there will be a frequency bias and $\text{AFSS} < 1$. It should be noted that because zero values outside of the domain are used in the convolution to calculate values at the edges of the domain, contrary to what is implied in Roberts and Lean (2008), the FSS is not always monotonically increasing and can obtain values somewhat larger than the AFSS when $\text{AFSS} < 1$ and $n < 2N - 1$.

Figure 13 shows the AFSS for RunM and RunA as a function of time. Figures 14 and 15 show FSS for RunM and RunA
with and without bias correction at four different forecast time periods. A threshold of 0.2 mg m$^{-2}$ was utilized. The score for a uniform and random forecast is also shown for comparison (Roberts and Lean, 2008).

The bias correction has a significant impact on FSS, mainly by decreasing the frequency bias and thus increasing the AFSS. A bias correction procedure which does not allow addition of positive values (right column) produced significantly better FSS scores for the NEP and ensemble mean than one that did allow addition of positive values (middle column) at larger scales,
but slightly worse FSS scores at the smaller scales. Allowing the addition of positive values in the bias correction led to more spread in the FSS values of the individual ensemble members.

As expected, skill decreases with time. The scale at which the NEP became greater than the uniform forecast in the third column reached about 2.5 degrees for the 09–10 UTC forecast. In agreement with (Ma et al., 2018), we found the NEP was more skillful than the ensemble mean.

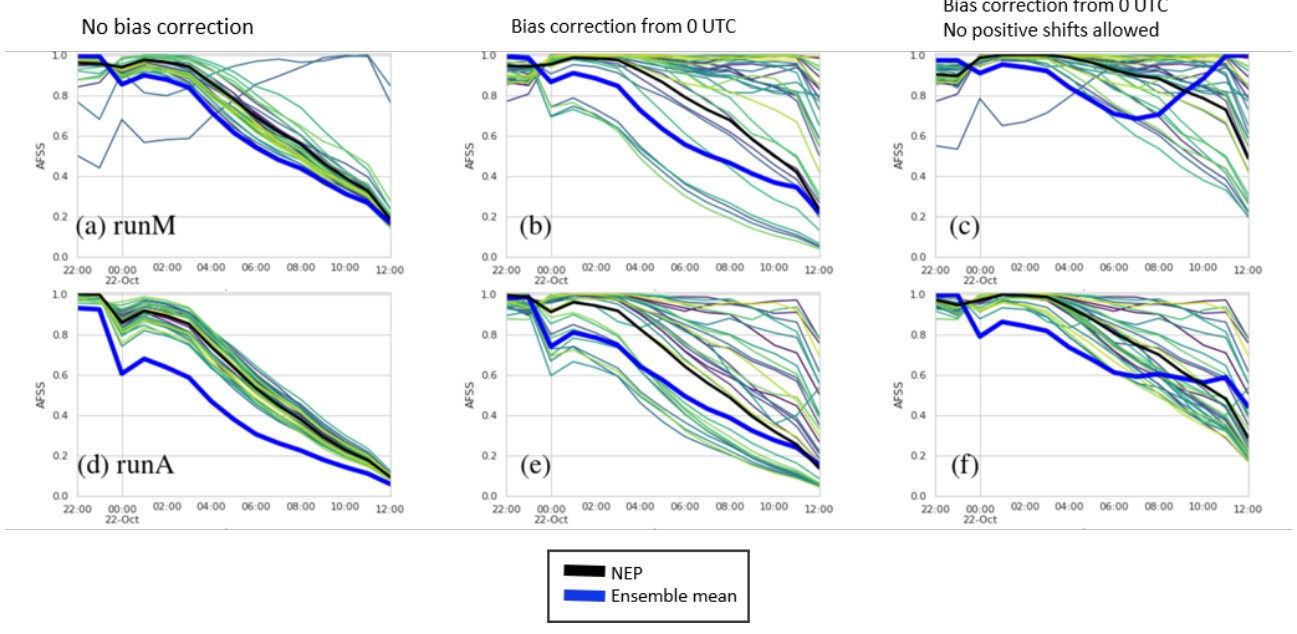

**Figure 13.** AFSS. Top row shows runA. Bottom row shows runB. Each column shows FSS for a different bias correction indicated by the text at the top of the column. Thin lines not shown in the legend are individual ensemble members. The time is for the beginning of the 1 h forecast time period.

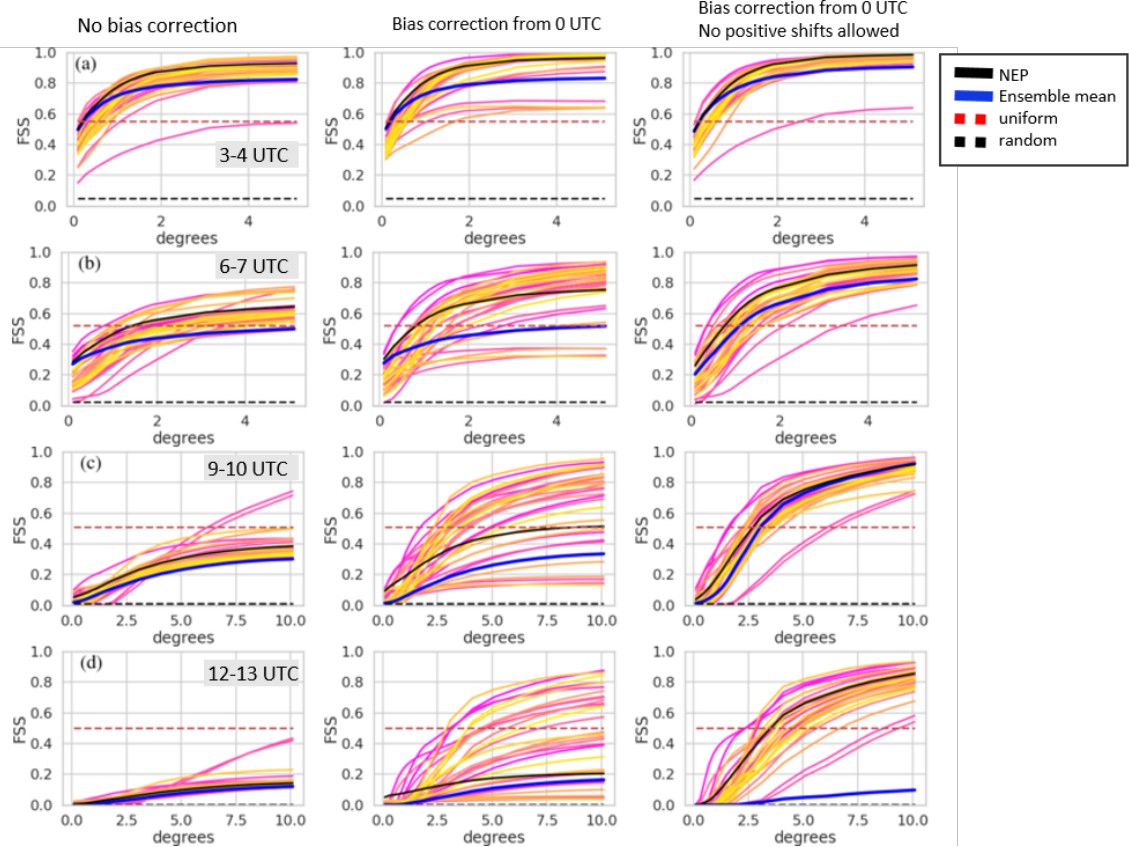

**Figure 14.** FSS vs. neighborhood size for RunM. Yellow, orange, pink lines not shown in the legend are results for individual ensemble members. Each row shows FSS for a different time period on 22 October 2020 indicated in the gray box. Each column shows FSS for a different bias correction indicated by the text at the top of the column. Note the change in scale on the x axis for rows (c) and (d).

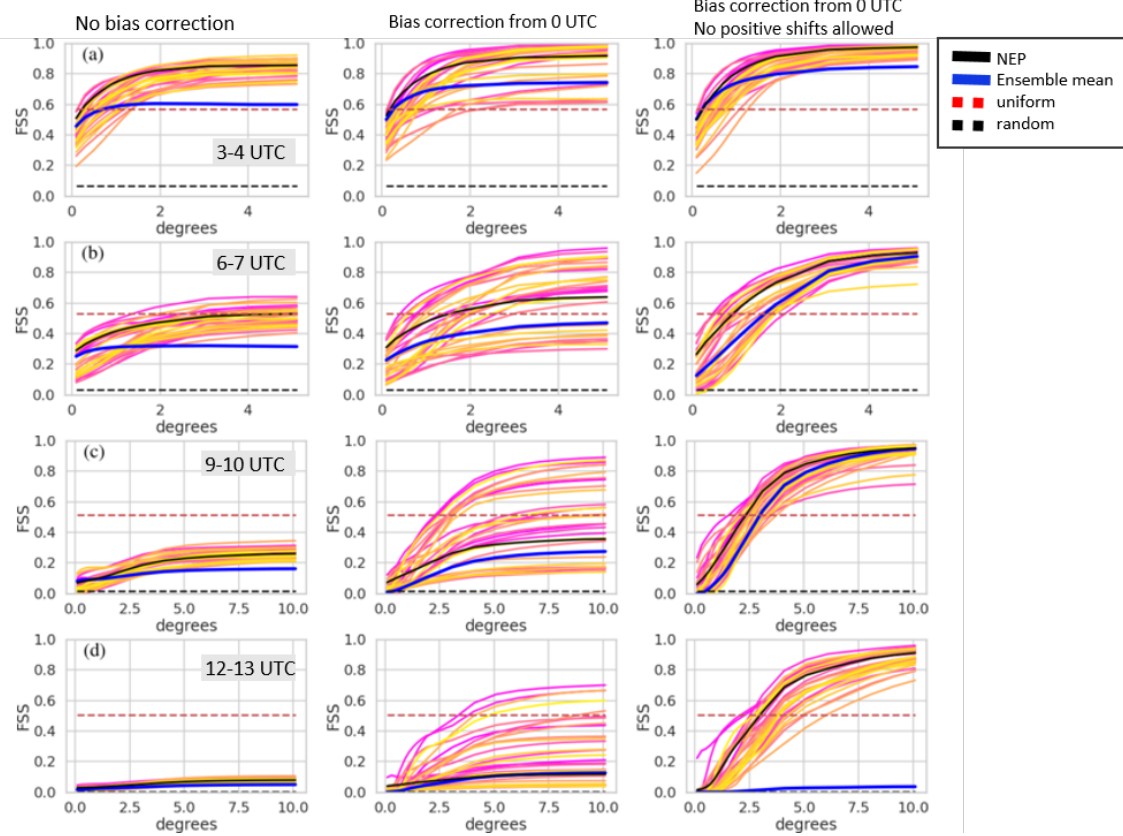

**Figure 15.** FSS vs. neighborhood size for RunA. Yellow, orange, pink lines not shown in the legend are results for individual ensemble members. Each row shows FSS for a different time period on 22 October 2020 indicated in the gray box. Each column shows FSS for a different bias correction indicated by the text at the top of the column. Note the change in scale on the x axis for rows (c) and (d).

  **6.7    Precision recall curve**

Many evaluation measures in use are based on a $2 \times 2$ contingency table in which the joint distribution of observations and forecasts can be split into four groups (Barnes et al., 2009; Wilks, 2011a; Miao and Zhu, 2021). Number of occasions, $a$, both the forecast and observation are above threshold; $b$, the forecast is above threshold but the observation is not; $c$, the forecast is not above threshold but the observation is; $d$, neither the observation or forecast is above threshold. $y_1$ denotes that the forecast is above threshold, while $y_2$ indicates it is not. $o_1$ indicates an observation above threshold while $o_2$ indicates an observation below threshold. This section will discuss the precision, probability of detection (POD), probability of false detection (POFD) and probability of false alarm, (POFA). We follow Barnes et al. (2009) for naming conventions of these quantities.

$$\text{precision} = P(o_1|y_1) = \frac{a}{a+b} \tag{12}$$

$$\text{POD} = P(y_1|o_1) = \frac{a}{a+c} \tag{13}$$

$$\text{POFD} = P(y_1|o_2) = \frac{b}{b+d} \tag{14}$$

$$\text{POFA} = P(o_2|y_1) = \frac{b}{a+b} \tag{15}$$

Evaluation statistics which employ $d$, are generally heavily dependent on the domain chosen as the forecast usually consists of one or possibly a few clouds of ash, surrounded by a large ash free area. $d$ can be changed from a small to a large number simply by cropping the domain close to the cloud or including a large area around the cloud. For instance, a measure such as POFD is highly sensitive to the domain size. The BS is sensitive to domain size, but skill scores, such as the BSS, discussed earlier are not sensitive to domain size because they utilize a reference forecast with the same domain size.

The following steps are used to create contingency tables for the probabilistic forecasts. First the ATL field is computed for a given mass loading threshold. The ATL field consists of values from 0 to 100 %. To convert the ATL field to a binary field, a probability threshold is applied as shown in Figure 9(c). Alternatively one could start with the APL field shown in Figure 9(e) and then apply the mass loading threshold to arrive at Figure 10(d).

The receiver operating characteristics (ROC) is a commonly used graphical forecast verification tool which plots POD vs. POFD for various probability thresholds applied to a probabilistic forecast (Wilks, 2011a; Saito and Rehmsmeler, 2015; Miao and Zhu, 2021). Generally speaking as the probability threshold increases, both POD and POFD decrease. However, some care must be taken in the interpretation of the curves as POFD which is plotted on the x axis of the ROC curve (not shown), is dependent on $d$ and thus can be increased or decreased by changing the size of the domain.

The Precision-Recall Curve (PRC) is a more appropriate evaluation tool (Saito and Rehmsmeler, 2015; Miao and Zhu, 2021) than the ROC as it does not depend on $d$. The PRC curve plots the precision vs. the POD. Although the curves tend to be more complicated than the ROC, the interpretation is still relatively straightforward.

The area under the curve (AUC) for either the ROC or PRC can be utilized to compare different forecasts. For both cases, an area closer to 1 is indicative of a better forecast.

Precision recall curves are shown in Figure 16. Points toward the right side of the curve are for low probability thresholds. The first point on the right is a 0 % probability threshold or uniform forecast in which all points in the scene are assumed to have some probability of ash. In this case the POD=1 and the precision is equal to the fraction of ash in the scene. The value of this point is also considered the baseline. When the area under the PRC curve is larger than the baseline value, the forecast skill is better than a uniform forecast. Note that the baseline value is dependent on the domain size.

As probability threshold increases, the POD either stays the same or decreases while the precision can either increase or decrease. For this case precision tends to increase sharply and then level off while POD decreases. In addition, the area under the PRC curve drops rather quickly with increasing forecast time. This is due mainly to a decrease in precision but decrease in POD is also a significant factor. For the time period from 09–10 UTC, a low probability threshold of $\frac{1}{31} = 3$ % does produce a POD of 1 indicating complete overlap with the observations. However, precision is also very low at this point, around 5 %. Increasing the probability threshold further causes the POD to drop below 20 %. The baseline value also decreases with forecast time largely because the domain increases. We crop the scene in a rectangular area that will include all ensemble members as well as the observation.

The bias correction has only a small effect on the PRC curve. It tends to increase precision, mostly at higher probability thresholds. For the bias correction shown in the figures which allows positive shifts, the POD is sometimes improved at low probability thresholds due to a few ensemble members with positive shifts covering a much larger area. If no positive shifts are allowed in the bias correction, then the POD is generally decreased.

The bias correction on the individual ensemble members is not effective at improving the PRC curve because it does not reduce the spatial spread of the entire ensemble. While the area covered by individual ensemble members is significantly decreased for many of the ensemble members by the bias correction, the area covered by the lower probability thresholds does not change much.

What does improve the PRC curve is using a coarser spatial scale. In order to look at different spatial scales, we utilize the NEP which was introduced in Sect. 6.6. To calculate the PRC curve, the NEP with various probability thresholds is compared to the observed field as illustrated in Figure 17. First, a mass loading threshold is chosen, here 0.2 g m$^{-2}$. The observed field is converted to a binary field (Figure 17(a)) and the simulated field is converted to ensemble frequency of exceedance (Figure 17(d)). Then a convolution between these fields and a square kernel of size n, the neighborhood size, is performed (Figure 17(b) and (e)). Finally the simulated field is converted back to a binary field by application of successive probability thresholds. The observed field is converted back to a binary field by setting any pixel that is above $\frac{1}{n^2}$ to one. This expands the observed field to all pixels within a distance n of the original field, the rational being that these pixels have a 100 % probability

of ash within a distance n. The fields shown in Figure 17 (c) and (f) are then used to compute the POD and precision for the PRC curve.

This method has some advantages over simply using a spatial average of the observed and modeled fields. First, we saw that the NEP performed fairly well as measured by the FSS. Secondly, spatial averaging will tend to decrease mass loadings especially for the small areas of ash considered in this case and would result in large areas and at some time periods, the whole observed and/or simluated cloud being below the 0.2 g m$^{-2}$ threshold. The method considered here provides information on how well the ensemble can predict when ash is above threshold within a neighborhood of size n.

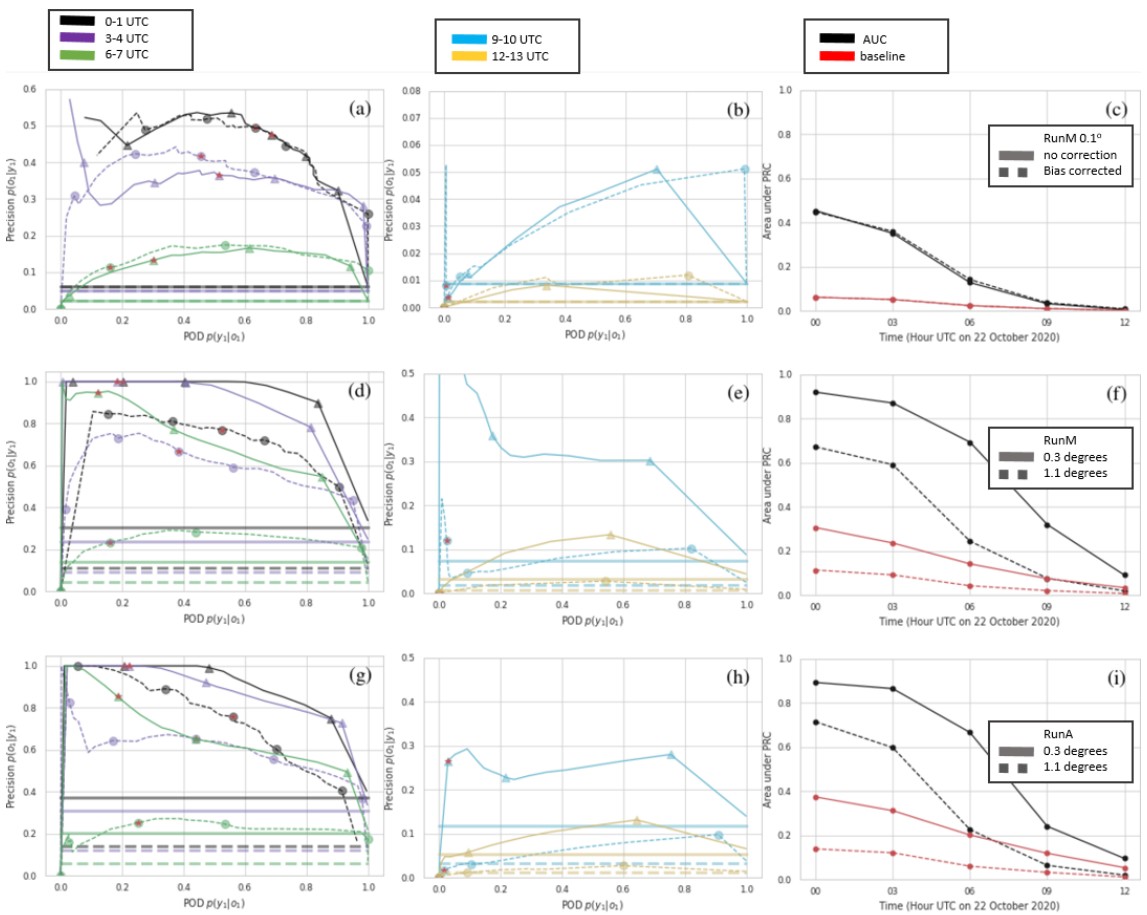

**Figure 16.** Top row (a,b,c) is RunM comparing bias corrected to non-bias corrected at the original spatial resolution. Second row (d,e,f) is RunM with bias correction and two different neighborhood sizes. Third row (g,h,i) is the same as the second row but for RunA. First and second columns show PRC for different time periods. Third column compares AUC and baseline. Precision and POD are computed for every probability threshold possible. For clarity symbols are only shown for probability thresholds of 3 %, 26 %, 48 %, 74 %, and 94 % (corresponding to 1,8,15,23,29 members out of 31 predicting the event). Additionally a red star is placed on the symbol representing the 48 % probability threshold.

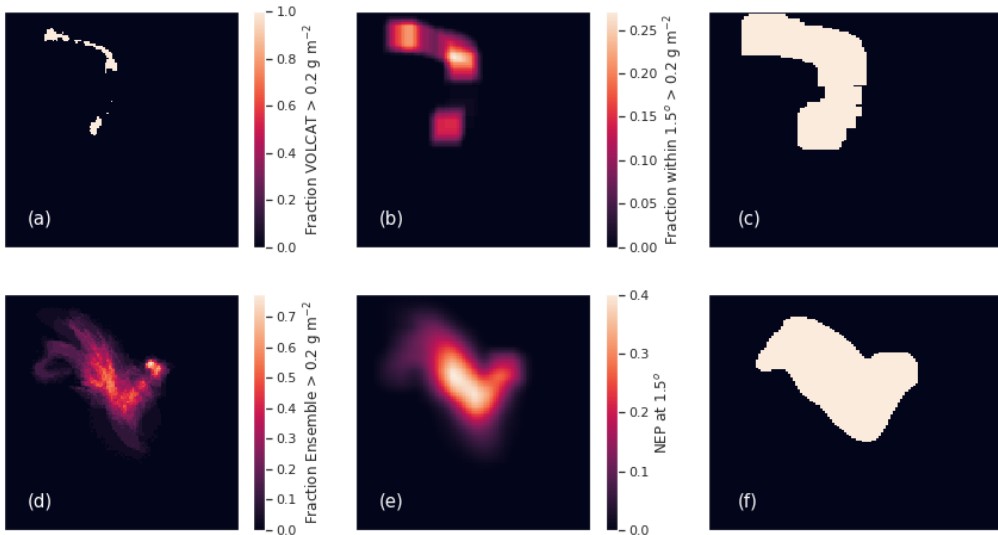

**Figure 17.** (a) Composited VOLCAT observations above 0.2 g m$^{-2}$. (b) VOLCAT observations from (a) convolved with a square kernel of size $15 \times 15$ pixels ($1.5^o$). (c) Pixels from (b) with values above $\frac{1}{15 \times 15}$. (d) ensemble relative frequency of exceedance of 0.2 g m$^{-2}$. (e) The NEP at $1.5^o$ spatial scale, i.e. values from (d) convolved with the square kernel. (f) values from (e) which are above $\frac{2}{31} = 0.06 = 6 \%$ .

Note that Folch et al. (2022) utilize what they call a generalized POD, POFA, precision, and Figure of Merit in Space (FMS). Folch et al. (2022) refers to POFA as false alarm rate, but this terminology can be confusing as it has been used to describe both equation 14 and equation 15 (Barnes et al., 2009). These quantities could be computed for precision or any quantity computed from the contingency table. Their use may be simpler for comparing different forecasts. On the other hand, we find the PRC curve to concisely convey information that is relevant to decision makers.

# 7  Conclusions

The flow field stretches and folds the ash cloud into complex three dimensional shapes, the exact placement and shape of which can be quite difficult to predict. Due to the chaotic nature of the flow field, a probabilistic approach is necessary. As there is little turbulent mixing high in the atmosphere, these areas are likely to have a high concentration gradient and thus fairly high concentrations of ash may exist within a small area. The location of those areas may be highly uncertain and to obtain a high POD, a low probability threshold for the ensemble frequency of exceedance may be needed. A low precision may be the trade-off. However, if dosage rather than concentration is the relevant quantity, then ensemble relative frequency of exceedance or ATL or APL should not be utilized, but instead probability of exceeding the dosage should be made from dosage calculation for each ensemble member. If dosage is the relevant quantity then predicting the exact location of small areas of ash may be less important than predicting their extent.

We utilized one simple case that is representative of common medium sized eruptions in the Northern latitudes with the goal of developing a workflow which includes source determination, bias correction, and forecast verification for probabilistic forecasts of ash for aviation. The workflow could be relevant for other applications in which gridded observations of the entire pollutant cloud are available.

When satellite retrievals of column mass loading are available, an inversion algorithm to determine height and time resolved emissions above the vent is an effective method of improving the forecast. In agreement with Chai et al. (2017), we found that utilizing all clear sky pixels in the inversion, tended to depress emission estimates due to transport errors in the simulation. However, utilizing no clear sky pixels, tended to produce emissions estimates which were not as realistic or stable. For instance, the emissions peak could be at the earliest time considered and was more dependent on the time range of the observations used. Using clear sky pixels which were several pixels away from the observations allowed for some transport error while removing from consideration emissions which produced areas of ash well removed from the observations.

This case illustrated the effectiveness of the CDF matching bias correction technique. The method is simple and fast, does not rely on spatial overlap between the simulated and observed fields, and can be utilized to improve the short term forecast. One factor left to be determined is whether allowing positive shifts in the forecast values is desirable. Allowing positive shifts can increase the POD, but may increase bias in some ensemble members at later forecast times and decrease the ensemble skill at larger neighborhood sizes (see Sect. 6.6). The technique as presented here assumes that it is desirable for the model to reproduce the observed CDF and does not consider possible errors in the observed CDF. The CDF matching is not affected by errors which do not change the CDF of the observations e.g. errors with Gaussian or uniform distribution and zero mean. However it is not unusual for mass loading retrievals to miss parts of the ash cloud due to meteorological cloud cover which could result in bias in the observed mass loadings. If the model bias is potentially very large, as it was for RunA initially, then utilizing the CDF matching with biased observations may still result in a smaller model bias. Ideally any bias in the observations would be corrected first. This could possibly be done by identifying areas which may be covered by meteorological cloud as well as identifying the shape the observed CDF should take.

The fit from the CDF matching may also simply be used for identifying how far apart the modeled and observed CDF's are in a similar fashion to the Kolmogorov-Smirnov (KS) parameter that has been used (Stein et al., 2015a; Crawford et al., 2016).

We introduced a suite of verification measures specifically for probabilistic forecasts of volcanic ash. The FSS was used to evaluate both the ensemble mean and NEP and indicates the spatial scale at which the ensemble has skill while the AFSS measures bias. The NEP which combines ATL or ensemble relative frequency of exceedance with probability of finding ash within a neighborhood, n, performs better than the ensemble mean. The rank histogram and reliability diagram provide information on ensemble consistency, calibration, and sharpness. The PRC curve provides POD and precision for different probability thresholds, information that may be used in risk-based approaches to flight planning.

Verification was performed on forecasts which utilized observations up to 00 UTC on 22 October 2020 for the source determination and observations up to 01 UTC for the bias correction. Time periods for verification were through the 12–13 UTC forecast time at which point the observed ash cloud covered less than 100 pixels of size $0.1^o \times 0.1^o$. Bias increased with forecast time. The spatial scale at which the forecasts showed skill increased with time. Reliability, resolution and sharpness of the probabilistic forecast also decreased with forecast time.

This work paves the way for future investigations and development. We found evidence similar to the findings of de Leeuw et al. (2021)and Cai et al. (2021) that modeled dispersion of the ash cloud does not match the observations. Further work should investigate possible reasons for this including revisiting model turbulence parameterisations for high altitudes.

Considerable work may be done to improve the construction of the ensemble. For instance, using an ensemble reduction or weighting technique (Stein et al., 2015b; Zidikheri et al., 2018) may be able to improve the ensemble performance. The verification measures introduced here are suitable for understanding how such techniques affect ensemble performance.

*Code and data availability.* HYSPLIT code is available at https://ready.noaa.gov/HYSPLIT.php. Code which was used to create the HYS-PLIT runs and post-process the results as well as data is available from a github repository https://github.com/noaa-oar-arl/hysplit_asheval_notebooks. Data in the form of meteorological files, HYSPLIT outputs, and satellite retrievals are available upon request.

## Appendix A: Meteorological conditions

Profiles of some of the model winds and temperature as well as time series of PBLH and precipitation are shown in Figure A1. The temperature profile indicates that the tropopause is around 300 mB which is approximately the height of injection estimated from the inversion algorithm in Sect. 4.3. Wind shear at and below this altitude is strong with wind speeds reaching a strong peak around 350 mb for the GFS and most GEFS ensemble members and wind direction also shifting from about 150 degrees to 300 degrees over about 5 km height diference. The inversion algorithm should be more accurate in conditions of strong wind shear. The height of the PBLH is not particularly important for this case as ash is injected well above this height.

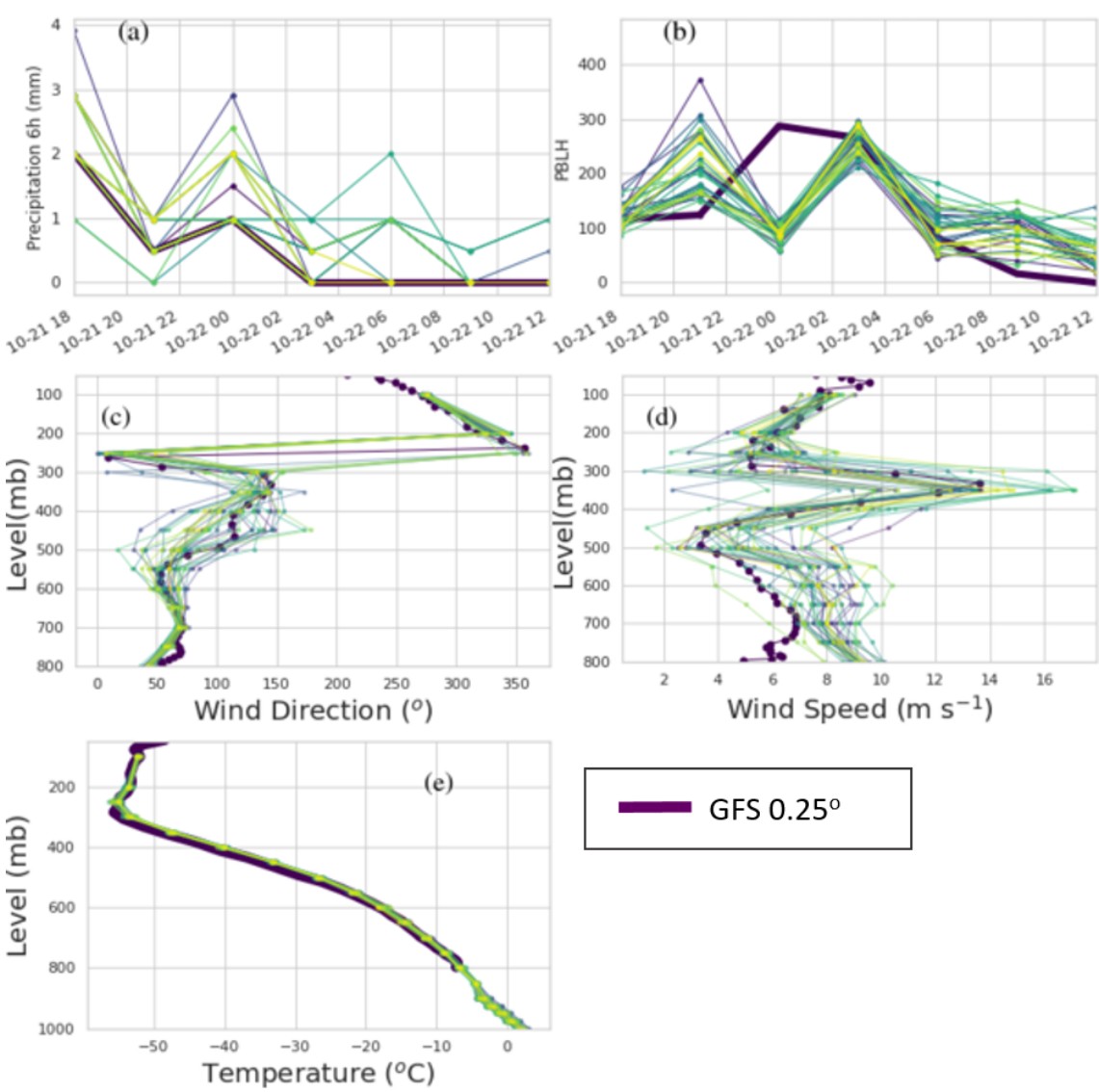

**Figure A1.** Time series of precipitation (a) and PBLH (b). Vertical profiles of wind direction (c), wind speed (d), and temperature (e) on 10 October 2020 at 21:00 UTC. Thin lines not shown in the legend are the different members of the GEFS.

## Appendix B: Particle Separation

Three HYSPLIT runs were performed with all inputs identical except for the particle size. Particle diameters of 0.6, 6, and 20 $\mu$m were used respectively. The particle positions were output once every hour. A Gaussian Mixture Model, GMM, consisting of 10 Gaussians was fit to the output at each hour as described in Crawford (2020). The score, $S_{ij}$, which is the per-sample average log-liklihood of the fit was calculated for the fits at each time. $i$ indicates the particle positions that the fit was made for and $j$ indicates the particle positions that the score was calculated for. Figure B1 shows $S_{iJ}$ as a function of time. The lines for

$i = j$ can be used as a reference. When the score for $i \neq j$ drops below the score for $i = j$, the fits for the two particles sizes can be said to be significantly different. We see that the position of the $20\mu$m particles becomes significantly different than those of the $6\mu$m particles at about 22 October 2020 00 UTC. The positions of the $0.6\mu$m particles do not become significantly different than those of the $6\mu$m particles during the period of interest. Figure B2 illustrates the separation of the 20 and 6 $\mu$m particles. As might be expected, the position of the larger particles is shifted downwards. Additionally there are far fewer particles at

latitudes below $49^o$.

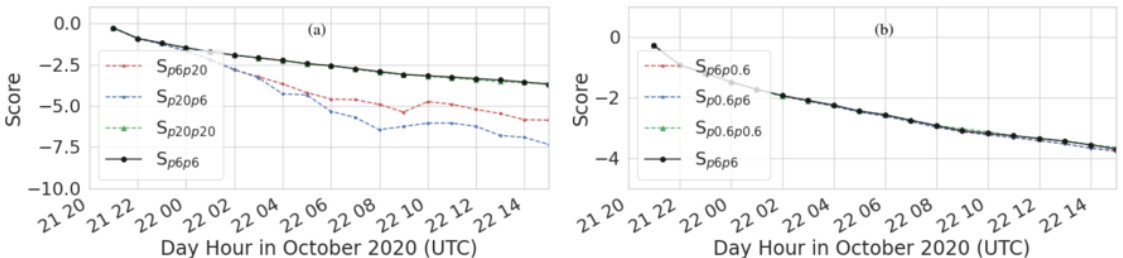

**Figure B1.** Score, $S_{ij}$, as a function of time. p0.6, p6, and p20 indicates 0.6, 6 and 20 $\mu$m particle positions respectively. (a) Comparison of scores for 6 and 20 $\mu$m size particles. (b) Comparison of scores for 0.6 and 6 $\mu$m sized particles.

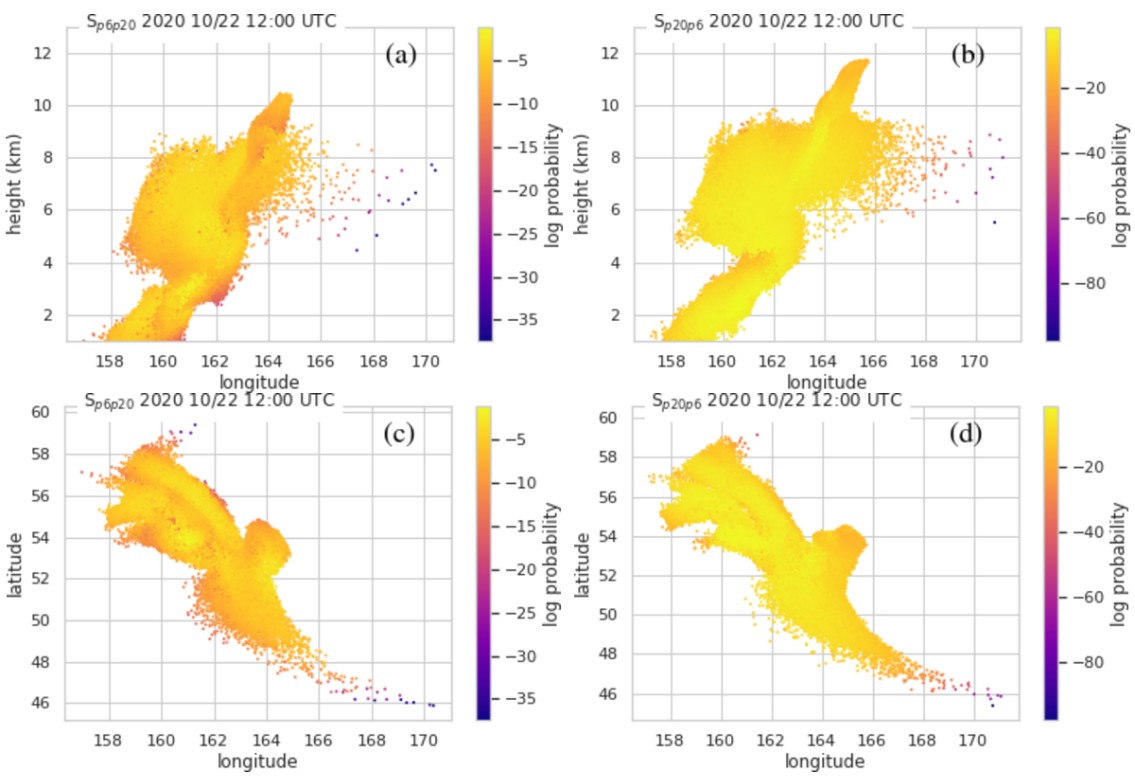

**Figure B2.** Comparison of computational particle positions for 6 and 20$\mu$m diameter particles at one time period. (a) and (c) show particle positions of 20$\mu$m size particles. (b) and (d) show particle positions of 6$\mu$m sized particles. The colors show log liklihood that the particle belongs in the fit that was created for the other sized particle. Thus darker colors indicate particles which have more displacement.

## Appendix C: Particle Number

The basic output of a Lagrangian particle model is the position of computational particles and the amount of mass each represents. This information is transformed into a concentration field by density estimation. HYSPLIT utilizes a simple bin counting density estimation scheme in which the total mass in the user defined bins is found by summing over the residence time weighted mass of each particle in the bin. Then time averaged concentration is arrived at by dividing by the volume. Although this scheme generally requires more particles than others, it has the advantage that the number of particles needed for a simulation can be estimated in a straighforward way for many model configurations (Crawford, 2020).

### C1 Particle number for inversion

We suppose that the quantity of 0.1 g m$^{-2}$ should be represented by at least 10 computational particles. Then with a horizontal resolution of $0.1^o \times 0.1^o$ at about $54^o$ latitude, the amount of mass on each computational particle should be no larger than $7.24 \times 10^5$ g. Therefore the total mass of each emission chunk which is represented by $2 \times 10^4$ particles should be no larger than about 0.0145 Tg. This condition is generally satisfied for this case.

To test we created runs identical to runB but with $1 \times 10^5$ particles per emission chunk as well as $2 \times 10^3$ particles per emission chunk. Both of these runs produced almost identical emissions estimates, that is, Figure 4 is almost the same for these runs.

### C2 Particle number for RunA and RunM

For RunA, $2 \times 10^4$ particles were released over a 2 hour time period. With a mass eruption rate of $3.75 \times 10^3$ kg s$^{-1}$, a model time step of 5 minutes, and an averaging time of 1 h, the lowest mass loading that the the model can produce (from one particle spending one time step in a grid cell) is 0.0016 g m$^{-2}$. The mass loadings of interest are about 100 times this, so we conclude that the particle number is sufficient.

The situation for RunM is somewhat more complicated. For the individual runs for the inversion algorithm, $2 \times 10^4$ particles were used as described above for RunB. Then a run with emissions that vary in time and space was created from the emissions estimates. Currently HYSPLIT evenly distributes the number of particles in time and space so when the emissions are varying, the amount of mass on the computational particles also varies. This makes a simple calculation such as done above difficult. To be on the safe side and because we did not have time constraints on the runs we ran with more than $2 \times 10^6$ particles total. The exact number varied for different ensemble members because of the way we handled emission chunks with essentially 0 emissions.

*Author contributions.* AC completed all the model runs, designed and performed the analysis and wrote the first draft of the paper. TC provided expertise and software for inverse modeling. BW, BS, and CPL provided the GEFS data in format suitable for ingest into HYSPLIT. BW assisted with the statistical analysis. AR contributed to code to process the observations and perform the statistical analysis. JS and MP

provided satellite retrievals and expertise on their use. BS and MP provided expertise on forecasting for aviation. MP helped secure financial support for the work. All authors provided review and editing of the manuscript.

*Competing interests.* The authors declare that they have no conflict of interest.

*Acknowledgements.* We would like to thank Jamie Kibler and Jeff Osiensky for providing information on VAAC operations. We thank Arnau
Folch and the anonymous reviewer for their careful review of the paper. Their feedback greatly improved the readability and conclusions of the paper.

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
