# Peer review of "Evaluation and Bias Correction of Probabilistic Volcanic Ash Forecasts"

_EGUsphere, 2022_

## Referee Comment (RC1)

Comments on "Evaluation and Bias Correction of Probabilistic Volcanic Ash Forecasts" by Crawford et al.

**Summary**

This paper considers ensemble-based volcanic ash forecasts using the HYSPLIT model with 31 GEFS meteorological forecasts for the October 2020 Bezymianny eruption. The cumulative distribution function (CDF) matching is used to reduce ensemble bias, and different metrics (rank histograms, reliability diagrams, fractions skill score, and precision recall curves) are used for further forecast verification. Different sets of runs are considered with various source terms and particle sizes, e.g. setting the source as on the operational source setting (run A) or through inversion (runs B and M) considering also different assimilation periods.

**General comments**

1. I found interesting the use of the Cumulative Distribution Function (CFD) matching as a way to correct ash forecast bias. This is, to my knowledge, quite novel in the field. I have some questions/comments here (Section 5):

- It is unclear to me how the bias correction is effectively applied in each assimilation cycle. Is it a cell-wise correction? How mass load corrections are converted to concentrations (or to "particle" masses in your Lagrangian framework)?
- The CDF linear fit considers the difference between the model/observation pairs (i.e. the absolute error) as a function of the forecast value, with line intercept shifting mass loading values in the direction opposite the sign of the intercept. My impression is that using absolute errors does not correct evenly the concentrated and the diluted (distal) parts of the cloud. Instead, would make sense to consider relative errors for the linear fit? I am curious so see if this would affect the resulting bias corrections.
- It is obvious that the CDF does not inform about cloud location; two clouds may have exactly the same CDFs without any overlap. You mention (line 166) that model points with no observation pair are discarded. This is unclear. Does it mean that bias correction (i.e. "data assimilation") is applied only in the overlap regions (i.e. where both model and observations are non-zero)? Or do you simply mean that the observations domain can be smaller than the model domain but that you actually assimilate zero observation values? Please clarify.
- As opposed to other DA methodologies, your DA strategy essentially brings model to observations without considering any observation error. The authors know well that in many cases large portions of clouds can be obscured. Could you comment on this?

2. I understand that, for bias correction (DA), you bring model to the observations space, which in the case of himawari-8 implies ~2km pixel size. I missed some discussion about whether discretization issues may exist in Lagrangian models like

HYSPLIT. In other words, is 20.000 particles a number large enough to guarantee convergence of model loads, particularly in the distal and/or at the end of the simulation? I strongly suggest running a simulation (e.g. run A or M) with 40k particle and see how this affects results, similar to what authors did with particle size or source width. I refer not only to CDF but also to the metrics in 6.7

3. Section 6.5 is hard to follow; I lost the thread even after reading two or three consecutive times. In particular:
- Did not understand how the reliability plots are computed and what the vertical axes in Figures 11 and 12 (b,c) show. Please explain better.
- Related to the previous point, what do you mean by "probability of observing the event (line 310)? Do you have an ensemble of observations??
- Section 6.5.1 confused me. What is the purpose?

4. A Table summarizing all the metrics you use (and the range of their possible values) would help.

**Minor comments and typos**

Line 24: "has developed" → "have developed"?
Line 38: 9km a.s.l.
Line 68: In addition to these physical mechanisms, could dilution below detection threshold explain also part of this decrease?
Line 100:
Line 103: "This is expected to produce a better forecast than for instance using only one cycle"…this is to be checked. For example, by mixing several forecast cycles you may introduce inconsistency in the wind fields, something undesirable in Eulerian frameworks (may be not that bad for HYPSPLIT).
Line 111: a mass fraction of ash of 0.1?
Line 187: "right" → "left"?
Line 194: 0:00Z → 0:00 UTC
Line 197: "This indicates a possible issue with turbulence parameterization in the model which control the rate at which the plume disperses". This is true, but actually could be more complex as diffusion effects can actually mix with wind shear advecting differently as particles settle down. With passive (non vertically resolved) satellite observations it is impossible to distinguish the single contributions from these two effects.
Line 266, eq (1). Please make evident that C depends also on time.
Line 275: "information about the relationship between concentrations in adjacent grid cells for each member is not preserved". I do not understand this sentence. Do you mean that HYSPLIT does not output concentration at height levels at different periods?
Lines 275-285. Argument difficult to follow, please explain better. Why dosage cannot be computed individually for each ensemble member and then do your percentiles?
Line 310: "P(..), giving the probability of observing…"
Line 359: (b)?

Line 395: You can also cite Folch et al. (2022) here, where skill scores are generalized to probabilistic contexts.

Line 406: Figures → Figure

---

## Referee Comment (RC2)

*Line 1.* *"Satellite retrievals of column mass loading of volcanic ash are **incorporated** into the HYSPLIT transport and dispersion modeling system"*.

Does that mean that the procedures illustrated in the paper are now available with Hysplit?

*Line 7.* *"the end of life of the ash cloud"*.

What do you mean with "end of life". Do you refer to the settling of ash particles on the ground, or do you refer to ash concentration in the atmosphere getting lower than a fixed threshold?

*Line 10.* *"small pieces of ash"*.

What are the small pieces of ash? It is not clear if small refer to the size of tephra, or to small portion of the original ash cloud. If you are referring to a portion of the ash cloud, please change to "parts", also in the rest of the text, "pieces" is confusing.

Line 20. "However, within the next five years".

Please add a reference.

Line 38. "The resulting ash cloud reached a plume height of around 9km"

Whenever a plume height is given, it should be stated if it is above sea level or above the vent.

Fruthermore, according with Horvath et al. 2021 (https://doi.org/10.5194/acp-21-12207-2021), "At 20:50 UTC (Fig. 10f and g), the overshooting top of the eruption column reaches its maximum altitude of ~ 15.3 km according to the side view technique".

Can you comment this difference in the estimation of maximum column height?

Lines 42-44. "The resulting plume forms a complicated three dimensional structure as it is stretched and folded by the wind field over the course of less than a day. As shown later, the exact location and shape of these structures is difficult to forecast."

Throughout the manuscript, the use of the terms "plume" and "cloud" are sometime confusing. For example, in line 38, the term plume seems to be used for the volcanic column. Because of that, here it is not clear if the 3D structure of the plume refers to the column or the cloud. I think that a choice should be done at the beginning and the use of the terms should be clearly stated, and then it should be consistent throughout the paper.

Line 46. "most of the ash is drawn into an area of low pressure, the location of which is fairly easy to forecast. The end fate of these large ash clouds is similar to that of the Bezymianny cloud discussed here".

Here the text seems to suggest that a pressure gradient drives the ash trajectories, while it is the drag exerted by the wind that controls them.

Line 49. "path of ash parcels emerging from a low pressure area".

I don't understand what this means. How do ash particles emerge from low pressure areas?

Line 50. Before this paragraph, I would be useful and interesting to have more details on the event (VEI, mass eruption rate, total grain size distribution, observed concentrations, deposit distance, wind condition at the vent), in order to have an idea of the size of the eruption.

Line 52. "the dissipation of the ash cloud".

Maybe "dispersion" instead of "dissipation". I've never seen this term used for ash cloud by the volcanological community.

Lines 56-61. It is important to give more information about the satellite retrievals. It is difficult to understand how the total mass or the total area are computed without knowing what are the data used. What is the maximum/minimum size of particles detected? Are you using an estimated columnar content? Is there a threshold used for the detection? Is there an uncertainty associated with the detection? To better understand how the area is computed, it would also be useful to have here two satellite images at different times, with a contour delimiting the area.

Line 64. "as well as a mass fraction of fine ash of 0.1".

Please justify this value and add a reference.

Line 64. "this would result in a plume height of about 8.2 km".

Without knowing the wind condition, it is difficult to judge if the use of the Mastin 2009 relationship gives reliable results. Is the plume strong or weak?

Line 68. "the decrease in mass over time is due to physical processes such as dissipation due to dispersion".

Mass is not dissipated, it is always conserved. The local concentration could decrease because of dispersion, but this does not decrease the total mass.

Line 69. "gravitational settling, and wet and dry deposition".

Here it seems that settling and deposition are two different things. Is this correct?

Lines 80-84.

This paragraph needs more details on the way Hysplit was used. From my knowledge, Hysplit can be used with particles or puff and, accordingly with this choice, with different dispersion/diffusion model associated with turbulence. Also, the way ash concentration is computed depends on the choice of particles or puffs. In both the cases, have you performed a convergence analysis on the number of particles/puffs needed to obtain a stable output? Forthermore, being Hysplit a Lagrangian model (i.e. it does not solve directly for mass concentration), it should be explained how concentration is computed from particles/puff position.

Line 110. "plume width was 1 km".

Again, it is not clear here if with "plume" you are referring to the volcanic column or to the ash cloud. Please clarify. If you are referring to the volcanic column, it is not clear where you assume this width/diameter? Is it at the base or at the top? In general, the diameter grows a lot from the base to the neutral buoyancy level. It is also not clear how the puffs/particles are released for this RunA. Is it from a cylinder, from a line, from the lateral surface of a cylinder? Maybe this does not make

any difference, but in any case, if the reader wants to try to replicate the results, he/she needs to have all the information required.

Line 110. "Vertical mass distribution was uniform from the vent at 2.88 km to the plume height".

Is the mass distribution or the release of ash uniform? In both the cases, is this assumption justified? For small ash particles, most of the mass reach the neutral buoyancy level and it is released in the atmosphere at that height.

Line 111. "and a mass fraction of fine ash"

Please specify the value used.

Lines 112-114. "would result in an MER of …".

It is not clear if the MER refers to the eruption rate of fine ash only or if it is the total rate. In general, this term is used to the total rate, so a different use would be confusing. If it is already the total eruption rate, the values reported in these lines seems low for a 10km volcanic column.

Line 116. "The inversion algorithm".

Because Hysplit allows for inversion of trajectories, I think that a reader could get easily confused here. Please give some more details on the inversion procedure from Chai et al.

Line 118. "and 1 km in the vertical and area of above the vent"

Something is missing here.

Line 118. "The modeling system only consider ash passively advected by the wind".

Aren't you considering the gravitational settling?

Line 128. "with the mass distributed"

Please change "mass" to "mass release".

Line 143. "The difference between using 20 and 6 $\mu$m particles".

What is the difference in the settling velocity for the two sizes?

Line 148. "lower emissions"

Does "lower" refer to the height or to the rate?

Line 153. "we find that this is in large part due to the dispersion of the ash cloud not being adequately represented by the model"

Is this really a limitation of the model or a limitation or of the retrieval algorithm?

Line 162. "Bias correction with CDF matching"

Starting from this section, I really struggled to understand what has been done, mostly because the algorithms and techniques applied are not described with enough details to make then clear. For example, for the CFD matching, it is written that "model values and observed values" are sorted, but there is no mention to what are the values used. Are they probabilities, concentrations, pixel values, values averaged for all the pixels? I really have no idea. And the figure does not help, because there are no units on the x-axis. You need first to explain clearly for which parameter/variable you compute the CDF.

Line 167. "A linear fit is applied to the difference between the pairs as a function of forecast value. Although, Reichle and Koster (2004) use higher order fits, we find that a linear fit is adequate."

Maybe I looked at the wrong reference, but I can't find any mention of fit in the paper cited here. Please check.

Line 172. "among the number of ash layers present".

What do you mean with layers of ash?

Line 172. "mass loading values".

Usually the volcanological community use this for the deposit (i.e. loading on the ground).

Lines 187-190.

I would move these lines in the previous subsection.

Line 195. "Increasing the spread of the cloud at early times to more closely match observations, caused the spread of the cloud at later times to have a larger mismatch."

The umbrella cloud intrudes as a gravity current at it takes some time to reach the maximum upwind and crosswind spreading. Have you tried to increase the size with time?

Line 198. "The modeled horizontal dispersion of the cloud is too fast".

This could be also done to a release of ash particles from a too large vertical interval, coupled with vertical wind shear. What happens when a larger fraction of particles is released from the neutral buoyancy level? As previously commented, particles of size you are considering here should, in large part, reach the top of the column.

Line 215. "We make the assumption that verification of modeled column mass loading values can be used as a proxy for verification of forecast concentrations."

The distinction between the use of column of mass loading and concentration should have been done at the beginning, because in most of the previous sections it was not clear which of the two forecasts and observations were referring to.

Line 229. "output of ash column mass loadings shown in Figure 8"

The use of a linear color scale makes more difficult to compare the results. Please use a log scale. Also the choice of colors does not help.

Lines 233-234. "By 12 UTC, this line of ash has broken into three small pieces, one just to the east of the volcano, one to the south, and one to the northwest."

Are these parts at different heights?

Line 249. "An example is shown in Figure 9(c) which shows number of model runs exceeding a given mass loading threshold"

I would remove "an example", because the number is not normalized in figure 9(c).

Line 250. "84 %".

I think that here and in the rest of the text you should remove the space between the number and the percentage symbol.

Lines 259-261. "In later sections we will utilize measures such as the precision recall curve, PRC, to evaluate the ATL at various probability thresholds. Keep in mind that these statistics are the same for the APL with the caveat that the point for 5 % probability of exceedance threshold for the ATL becomes the point for the APL of 95 %."

This is difficult to understand, because the PRC has not been introduced so far in the manuscript.

Line 263. "Some sources"

Which sources? Add references.

Line 267. "If velocity is constant then D is not sensitive to spatial averaging that is performed parallel to the flight path."

This is true, but when computing the concentration in the grid cell, averaging is preformed both in directions parallel and normal to the flight path, so computed dosage depends on grid resolution.

Line 292. "Instead predicting the time at which the ash cloud breaks into small enough pieces to be ignored becomes important."

Written in this way it seems that it is common to observe the ash cloud breaking into small parts, but I'm not so sure it is always the case. It also not completely clear to me what "breaks into small pieces" means. Is it because between these "small pieces" ash concentration is very low and so it is not detected? Or is the ash cloud really splitting into different and isolated parts?

Lines 295-298. "The extent of lower mass loading of ash, 0.2 g m$^{-2}$ continues to increase through 12 UTC. In contrast, the extent of the higher mass loadings follows the observations more closely. This mismatch occurs because the spatial gradient in the observations is much steeper than in the simulation. This is again an indication that the modeled turbulent dispersion which controls the spread of the ash is not reproducing what is observed."

This is not clear to me. If the extent of the area exceeding a threshold is too large, that does not necessarily mean that there is problem with turbulent dispersion. In fact, when the bias correction is applied, it seems that the areas are better reproduced.

Lines 309-315.

I'm sorry but the description is the reliability diagram is extremely confusing to me. In the simulations you have the ensemble of simulations, the output at different times, the output at different pixels. When you write "the probability of observing the event" what do you mean? Is it the probability for a single element of the ensemble, considering all the pixels at a single time? Is it the probability associated with the variability in the ensemble elements? Is it the variability associated with the different output times in a time interval? I have the same problem with the variable plotted in the vertical axis of the middle column in Fig.11, what does "Fraction observations" mean? I have the same problem with the refinement distribution. "The second part of the diagram is the refinement distribution which is a histogram showing how often the modeled probability, $y_i$, occurred". You need to state more clearly what you mean with "modeled probability".

Section 6.5.2

In this section I have problems to understand the discussions on the reliability diagram and the refinement distribution because, as written above, it is not clear what is plotted in the figures.

Lines 379-380. "Observed and modeled fractions are then computed for different neighborhood sizes, n, by convolution of the field with a square kernel of that size".

I suggest to change to:

"Observed and modeled fractions are computed by convolution of the field with a square kernel, for different kernel sizes, n.".

Line 384.

Before equation 3, please define O(n) and m(n). I assume they are the fractions, but it would be better to state it explicitly.

Lines 389-391. "When computing the FSS, it is standard for the reference forecast to be defined as the largest possible MSE that can be obtained from the forecast and observed fractions".

Isn't this number just 1? By looking at Eq.3, you maximize the MSE when you maximize each addend of the sum, and this is obtained when, in the difference between O(n) and m(n), one value is 0 and the other is 1. I also don't understand why in Eq.5, with respect to Eq.3, the square moved inside the parenthesis and there is a difference instead of a sum.

Line 397. "At some value of n <= 2N-1"

What is N?

Line 419. "Number of occasions"

Here I have the same problem I had with the description of the reliability diagram. The definition of the parameters seems to ignore that you are applying the technique to a specific application, and it is given as an abstract definition. What does the term "occasions" mean? I don't understand if you refer to the number of pixel for a simulation of the ensemble, or to the number of ensemble for a pixel.

Line 431. "For instance, a measure such as F is highly sensitive to the domain size"

What is F here?

Line 435. "various probability thresholds".

Is this correct or should it be just "various thresholds", without probability? Is this the same threshold used to define a in Lines 419-420? "both the forecast and observation are above threshold". If it is really a probability threshold, please can you write explicitly to what the probability refers to?

Line 436-437. "However, some care must be taken in the interpretation of the curves as POFD which is plotted on the x axis".

Where is the x-axis you mention? There is no reference to any figure here.

Line 443-end of subsection.

I confess that I was lost here, because I could not understand clearly the probability thresholds, as written above. In the two first columns of Fig.16 there are also a lot of markers, but I can't understand what they represent. Probably there is something I missed.

Figures

Figure 1. Please add letters to the panels. In panel (a), is is e^(-xx), with e being the Euler's number, or 10^(-xx)? In Panel (d), please write if the height is a.s.l.

Figure 6. In panel b and d add a label to the vertical axis.

Figure 8. The color scale does not help to distinguish low concentrations from the background. I think it would also work better a log scale.

Figure 9. I think you should remark that the color scale here is not linear or log, but I think a mix of the two.

---

## Author Comment (AC1)

We thank Arnau Folch for the careful review and thoughtful comments which will improve the clarity of our manuscript. Some responses are posted in blue below the comment they correspond to. Comments which are not addressed here will be addressed in the final review period.

- It is unclear to me how the bias correction is effectively applied in each assimilation cycle. Is it a cell-wise correction? How mass load corrections are converted to concentrations (or to "particle" masses in your Lagrangian framework)?
- Since the bias correction is applied to the gridded mass loadings (or concentrations) the procedure would be the same for an Eulerian model. The procedure isn't applied to the computational particle masses but to the gridded mass loading field that was estimated form the mass distribution of the computational particles.
- It is a cell-wise correction. s' = (1-m)s - b where s' is the corrected mass loading for the cell, s is the original mass loading. m is the slope and b is the intercept.
- As for applying the correction to concentrations - Line 171 states "There are several practical considerations in adding or subtracting a constant value to the simulated mass loading or concentration values. Propagating the correction to ash concentrations would involve some assumptions such as dividing the additive correction evenly among the number of ash layers present". This can be seen because the mass loading at each cell, is the sum of the concentrations,c, over each ith level in the column, multiplied by the level thicknesses, L.
- $s = \sum_i LiCi$
- $s' = (1-m) \sum_i LiCi - b$
- Ci' = (1-m)Ci – bi
- $\sum_i bi = b$
  The corrected concentration Ci' can be calculated by multiplying the uncorrected concentration by (1-m) and then subtracting a fraction of the intercept, bi. The only constraint is that the sum of the bi equal to b. Different strategies could be devised for estimating the bi values. We could go into some ideas in more depth but it seems outside the scope of the paper.

- The CDF linear fit considers the difference between the model/observation pairs (i.e. the absolute error) as a function of the forecast value, with line intercept shifting mass loading values in the direction opposite the sign of the intercept. My impression is that using absolute errors does not correct evenly the concentrated and the diluted (distal) parts of the cloud. Instead, would make sense to consider relative errors for the linear fit? I am curious so see if this would affect the resulting bias corrections.
- Is the impression from Figure 6 (a) and (c) showing the line fit deviating from the actual differences at higher forecast values? This is not always the case and the fit can sometimes deviate more for lower values. To make the correction more even, a higher order fit may be used. It would also be possible to use a weighted linear regression to make the fit better for higher mass loadings at the cost of possibly making it worse for lower mass loadings. This could be desirable if certain ranges were more important for end users. Currently we use

y-x = mx + b. Where y is observation and x is model value. Utilizing (y-x)/x = mx +b would be equivalent to y-x = mx2 + bx. Which is equivalent to using a second order fit with the y intercept forced to zero. It is probably preferable to just utilize a second order fit.

- It is obvious that the CDF does not inform about cloud location; two clouds may have exactly the same CDFs without any overlap. You mention (line 166) that model points with no observation pair are discarded. This is unclear. Does it mean that bias correction (i.e. "data assimilation") is applied only in the overlap regions (i.e. where both model and observations are non-zero)? Or do you simply mean that the observations domain can be smaller than the model domain but that you actually assimilate zero observation values? Please clarify.

- The following re-wording of lines 165-166 may be more clear. "The modeled and observed values are sorted from greatest to least and then paired so the greatest observed value is paired with the greatest modeled value and so forth. Pairs in which either the observed or modeled value are 0 are discarded. Usually there are more modeled values above 0 than observed because modeled values can cover a larger range."

- As opposed to other DA methodologies, your DA strategy essentially brings model to observations without considering any observation error. The authors know well that in many cases large portions of clouds can be obscured. Could you comment on this?

- We will add some discussion this to the conclusions.

- First we note that CDF matching is not affected by errors which do not change the CDF. For instance, errors with Gaussian or uniform distribution and zero mean. Clearly it will be affected by errors that result in the observed CDF being different than the actual CDF such as bias that occurs when the retrieval fails for a portion of the cloud. The CDF matching will then result in model output with a bias close to that of the observations. We note that some other DA methodologies are bias-blind as well.

- If the model bias is very large, as it was for RunA, then it may still be useful to correct it using the incomplete observations as the observation bias might be significantly smaller than the model bias. However, for a run which has already assimilated observations in some way such as RunM where the model bias is expected to be fairly small, then using CDF matching with incomplete observations would probably not be useful.

- It may be possible to add in some accounting for incomplete observations by trying to correct the bias in the observations first. For instance identifying areas which may be covered by cloud as well as identifying what sort of shape the observed CDF/PDF should take. If the observed CDF can be shown to take a certain form that can be parameterized, then this could be possible.

2. I understand that, for bias correction (DA), you bring model to the observations space, which in the case of himawari-8 implies ~2km pixel size. I missed some discussion about whether discretization issues may exist in Lagrangian models like HYSPLIT. In other words, is 20.000 particles a number large enough to guarantee convergence of model loads, particularly in the distal and/or at the end of the simulation? I strongly suggest running a simulation (e.g. run A or M) with 40k particle and see how this affects results, similar to what authors did with particle size or source width. I refer not only to CDF but also to the metrics in 6.7

We actually bring observations to the model space by regridding to a 0.1x0.1 degree grid and time averaging over an hour. This is discussed in section 3, line 70. "For comparison to model output, the satellite data is parallax corrected using estimated cloud top heights and then composited by first regridding to a regular 0.1 degree latitude-longitude grid and subsequently taking the average of all retrievals within an hour time frame"

We agree that it is worthwhile to discuss the choice of particle number in more depth and will add an appendix with the following discussion. The lowest concentration or mass loading that a Lagrangian model can resolve with the method of estimation used here is determined by the grid size, time step, and amount of mass on the computational particle.

We suppose that the quantity of $0.1$ g m$^{-2}$ should be represented by at least 10 computational particles. Then with a horizontal resolution of 0.1o ×0.1o at about 54o latitude, the amount of mass on each computational particle should be no larger than
$7.24\times10^5$ g. Therefore the total mass of each emission chunk which is represented by $2\times104$ particles should be no larger than about 0.0145 Tg. This condition is generally satisfied for this case. To test we created runs identical to runB but with $1\times10^5$ particles per emission chunk as well as $2\times10^3$ particles per emission chunk. Both of these runs produced almost identical emissions estimates, that is, Figure 4 is almost the same for these runs.

For RunA, $2\times10^4$ particles were released over a 2 hour time period. With a mass eruption rate of $3.75\times10^3$ kg s$^{-1}$, a model time step of 5 minutes, and an averaging time of 1 h, the lowest mass loading that the the model can produce (from one particle spending one time step in a grid cell) is $0.0016$ g m$^{-2}$. The mass loadings of interest are about 100 times this, so we conclude that the particle number is sufficient. The situation for RunM is somewhat more complicated. For the individual runs for the inversion algorithm, $2\times10^4$ particles were used as described above for RunB. Then a run with emissions that vary in time and space was created from the emissions estimates. Currently HYSPLIT evenly distributes the number of particles in time and space so when the emissions are varying, the amount of mass on the computational particles also varies. This makes a simple calculation such as done above difficult. To be on the safe side and because we did not have time constraints on the runs we ran with more than $2\times10^6$ particles total. The exact number varied for different ensemble members because of the way we handled emission chunks with essentially 0 emissions.

3. Section 6.5 is hard to follow; I lost the thread even after reading two or three consecutive times. In particular:
- Did not understand how the reliability plots are computed and what the vertical axes in Figures 11 and 12 (b,c) show. Please explain better.
- Related to the previous point, what do you mean by "probability of observing the event (line 310)? Do you have an ensemble of observations??
- The addition of the following may answer the previous two points. "The modeled probability of the event on the x axis indicates the fraction of ensemble members which indicate the event occurred. If the modeled probability is 50% then we would

expect that if we look at all the times the modeled probability was 50%, half the time the event would be observed and half the time it would not be observed. The y axis gives the actual fraction of times the event was observed. Ideally, the calibration function lies along the 1:1 line. When the function lies below the 1:1 line, the modeled probabilities are overconfident. For instance a point at (0.80,0.50) means that out of all the times the model predicted there was an 80% chance of occurrence, the actual event was observed only 50% of the time."

- Section 6.5.1 confused me. What is the purpose?
- This may be more clear if the content is moved after or within 6.5.2. The purpose is to provide a more intuitive understanding of the temporal evolution of the rank histogram and reliability diagrams. Also to convey that simply having a flat rank histogram does not necessarily mean a great forecast. The section is referred to in lines 383-385. "As forecast time increases, the case with bias correction approaches, but does not reach, the simple case discussed earlier. The ensemble members overlap less and less with each other which is indicated in the refinement distribution. The calibration function becomes flat with all simulated probabilities corresponding to a low actual probability. The rank histogram becomes quite flat on the lower end indicating a large number of points with below threshold value for the observations and more than half the ensemble members. This is due to increasing difficulty in predicting the location of the ash."

4. A Table summarizing all the metrics you use (and the range of their possible values) would help.

**Minor comments and typos**

Line 24: "has developed" ➜ "have developed"?

Line 38: 9km a.s.l.

Line 68: In addition to these physical mechanisms, could dilution below detection threshold explain also part of this decrease?

Yes, this is what is meant by "dissipation due to dispersion." Wording will be updated to be more clear.

Line 100:

Line 103: "This is expected to produce a better forecast than for instance using only one cycle"…this is to be checked. For example, by mixing several forecast cycles you may introduce inconsistency in the wind fields, something undesirable in Eulerian frameworks (may be not that bad for HYPSPLIT).

This is a good point. Wording will be changed to reflect. Inconsistency in wind fields manifests different problems in a Lagrangian model vs. Eulerian. For instance,violation of mass conservation is not an issue in Lagrangian model. However violation of the well-mixed condition could occur which could result in spurious higher concentrations in some areas. This is usually a rather subtle effect and the benefits of more accurate wind speeds and directions would be expected to dominate. However this has not been investigated.

Line 111: a mass fraction of ash of 0.1?

Line 187: "right" ➜ "left"?

Line 194: 0:00Z ➜ 0:00 UTC

Line 197: "This indicates a possible issue with turbulence parameterization in the model which control the rate at which the plume disperses". This is true, but actually could be more complex as diffusion effects can actually mix with wind shear advecting

differently as particles settle down. With passive (non vertically resolved) satellite observations it is impossible to distinguish the single contributions from these two effects.

Line 266, eq (1). Please make evident that C depends also on time.

Line 275: "information about the relationship between concentrations in adjacent grid cells for each member is not preserved". I do not understand this sentence. Do you mean that HYSPLIT does not output concentration at height levels at different periods?

Lines 275-285. Argument difficult to follow, please explain better. Why dosage cannot be computed individually for each ensemble member and then do your percentiles?

We do state this in Line 268 "For probability of exceeding a critical dosage $D_C$, this equation should be applied to each ensemble member and ensemble relative frequency of exceeding the dosage computed from the resulting ensemble of dosages" The conversation here was meant to convey that if an end user does not have access to the full ensemble data, and only has APL or ATL data, then information on dosages cannot be accurately inferred from just that information. The following changes may convey this better (underlined is added).

"However,  utilizing the probability of exceeding a concentration or ATL to get probability of exceedance of dosage along a route containing multiple grid cells is not possible because information about the relationship between concentrations in adjacent grid cells for each member is not preserved by the ATL field.

---

## Author Response (AR2)

**1  Reply to Reviewers**

We appreciate the reviewers thorough review of the paper and have done our best to respond to all comments. Both reviewers found parts of the paper difficult to follow. To address this many parts of the paper were reworded, reorganized, or additional details were added.

Some highlights of the changes are

- An improved description and discussion of the CDF matching technique including additional references.

- An additional table summarizing the evaluation measures.

- An extensively reworked Section 6.5 on the Rank histogram and reliability diagram.

- An additional appendix which discusses how the number of computational particles used for simulations was chosen.

Replies to each comment can be found in the sections below. Reviewer comments are in black. Replies are in blue.

In addition we found and corrected the following error as we were making the corrections recommended by the reviewers.

- Most of the references for Wilks were meant to be for the book, Statistical Methods in the Atmospheric Sciences, and were accidentally attributed to an article, Sampling distributions of the Brier score and Brier skill score under serial dependence.

**1.1  Reviewer 1**

I found interesting the use of the Cumulative Distribution Function (CFD) matching as a way to correct ash forecast bias. This is, to my knowledge, quite novel in the field. I have some questions/comments here (Section 5):

- It is unclear to me how the bias correction is effectively applied in each assimilation cycle. Is it a cell-wise correction? How mass load corrections are converted to concentrations (or to "particle" masses in your Lagrangian framework)?

  - Since the bias correction is applied to the gridded mass loadings (or concentrations) the procedure would be the same for an Eulerian model. The procedure isn't applied to the computational particle masses but to the gridded mass loading field that was estimated from the mass distribution of the computational particles

  - It is a cell-wise correction. $s' = (1 - m)s - b$ where $s'$ is the corrected mass loading for the cell. $s$ is the original mass loading. $m$ is the slope and $b$ is the intercept.

  - As for applying the correction to concentrations - Line 171 states "There are several practical considerations in adding or subtracting a constant value to the simulated mass loading or concentration values. Propagating the correction to ash concentrations would involve some assumptions such as dividing the additive correction evenly among the number of ash layers present". This can be seen because the mass loading at each cell, is the sum of the concentrations,c, over each ith level in the column, multiplied by the level thicknesses, L.

$$s = \sum_{i}^{N} L_i c_i \tag{1}$$

$$s' = (1 - m)\sum_{i}^{N} L_i c_i - b \tag{2}$$

$$c_i' = (1 - m)L_i c_i - b_i \tag{3}$$

$$\sum_{i}^{N} b_i = b \tag{4}$$

The corrected concentration $C_i'$ can be calculated by multiplying the uncorrected concentration by $(1 - m)$ and then subtracting a fraction of the intercept, $b_i$. The only constraint is that the sum of the $b_i$ equal to $b$. Different strategies could be devised for estimating the $b_i$ values. We could go into some ideas in more depth but it seems outside the scope of the paper

These descriptions and equations were added to Section 5.

- The CDF linear fit considers the difference between the model/observation pairs (i.e. the absolute error) as a function of the forecast value, with line intercept shifting mass loading values in the direction opposite the sign of the intercept. My impression is that using absolute errors does not correct evenly the concentrated and the diluted (distal) parts of the cloud. Instead, would make sense to consider relative errors for the linear fit? I am curious so see if this would affect the resulting bias corrections.

  The point that other methods of fitting may work better is a good one. We have improved the introduction to the CDF matching in Section 5. We added some references and also included the following "A linear fit is applied to the difference between the pairs as a function of forecast value. Higher order fits or other functional forms can be used or a non-parametric approach sometimes referred to 200 as quantile mapping can be used (Piani et al., 2010; Gudmundsson et al., 2012). Future work may include comparing different methods of constructing the transform function."

  Is the impression from Figure 6 (a) and (c) showing the line fit deviating from the actual differences at higher forecast values? This is not always the case and the fit can sometimes deviate more for lower values. To make the correction more even, a higher order fit may be used. It would also be possible to use a weighted linear regression to make the fit better for higher mass loadings at the cost of possibly making it worse for lower mass loadings. This could be desirable if certain ranges were more important for end users. Currently we use $y - x = mx + b$. Where $y$ is observation and $x$ is model value. Utilizing $(y - x)/x = mx + b$ would be equivalent to $y - x = mx2 + bx$. Which is equivalent to using a second order fit with the $y$ intercept forced to zero. It is probably preferable to just utilize a second order fit.

- It is obvious that the CDF does not inform about cloud location; two clouds may have exactly the same CDFs without any overlap. You mention (line 166) that model points with no observation pair are discarded. This is unclear. Does it mean that bias correction (i.e. "data assimilation") is applied only in the overlap regions (i.e. where both model and observations are non-zero)? Or do you simply mean that the observations domain can be smaller than the model domain but that you actually assimilate zero observation values? Please clarify.

  The following re-wording of lines 165-166 may be more clear. "The modeled and observed values are sorted from greatest to least and then paired so the greatest observed value is paired with the greatest modeled value and so forth. Pairs in which either the observed or modeled value are 0 are discarded. Usually there are more modeled values above 0 than observed because modeled values can cover a larger range."

- As opposed to other DA methodologies, your DA strategy essentially brings model to observations without considering any observation error. The authors know well that in many cases large portions of clouds can be obscured. Could you comment on this?

  – We have added some discussion of this to the conclusions.
  – First we note that CDF matching is not affected by errors which do not change the CDF of the observations. For instance, errors with Gaussian or uniform distribution and zero mean. Clearly it will be affected by errors that result in the observed CDF being different than the actual CDF such as bias that occurs when the retrieval fails for a portion of the cloud. The CDF matching will then result in model output with a bias close to that of the observations. We note that some other DA methodologies are bias-blind as well.
  – We have added more references for CDF matching, Gudmundsson et. al. 2012 and Piani et. al. 2010. These sources utilize the method as a post-processing on climate models. The CDF matching isn't intended to be used instead of DA methods but can be utilized

in addition to other DA methods. We also add Belitz. et. al. 2021 which looks at the technique as a post-processing for machine learning models.

– If the model bias is very large, as it was for RunA, then it may still be useful to correct it using the incomplete observations as the observation bias might be significantly smaller than the model bias. However, if the model bias is expected to be fairly small or a conservative forecast with a large bias is preferable to one with a low bias, then using CDF matching with incomplete observations may not be useful.

– It may be possible to add in some accounting for incomplete observations by trying to correct the bias in the observations first. For instance identifying areas which may be covered by cloud as well as identifying what sort of shape the observed CDF/PDF should take. If the observed CDF can be shown to take a certain form that can be parameterized, then this could be possible.

I understand that, for bias correction (DA), you bring model to the observations space, which in the case of himawari-8 implies 2km pixel size. I missed some discussion about whether discretization issues may exist in Lagrangian models like HYSPLIT. In other words, is 20.000 particles a number large enough to guarantee convergence of model loads, particularly in the distal and/or at the end of the simulation? I strongly suggest running a simulation (e.g. run A or M) with 40k particle and see how this affects results, similar to what authors did with particle size or source width. I refer not only to CDF but also to the metrics in 6.7

- We actually bring observations to the model space by regridding to a $0.1^o \times 0.1^o$ grid and time averaging over an hour. This is discussed in section 3. *"For comparison to model output, the satellite data is parallax corrected using estimated cloud top heights and then composited by first regridding to a regular $0.1^o$ latitude-longitude grid and subsequently taking the average of all retrievals within an hour time frame"*

- We agree that it is worthwhile to discuss the choice of particle number in more depth and have added an appendix which does so. The lowest concentration or mass loading that a Lagrangian model can resolve with the method of estimation used here is determined by the grid size, time step, and amount of mass on the computational particles.

3. Section 6.5 is hard to follow; I lost the thread even after reading two or three consecutive times. In particular:

- Did not understand how the reliability plots are computed and what the vertical axes in Figures 11 and 12 (b,c) show. Please explain better.

  We re-wrote our explanation of the reliability diagrams. As part of this re-writing, we also relabeled the axis in Figures 11 and 12 (b,c).

- Related to the previous point, what do you mean by "probability of observing the event (line 310)? Do you have an ensemble of observations??

  We re-wrote this section.

- Section 6.5.1 confused me. What is the purpose? We no longer divide Section 6.5 into two subsections. We combined the content in both sections and moved the content in Section 6.5.1 which was previously at the beginning, near the end of the section. The purpose is to provide a more intuitive understanding of the temporal evolution of the rank histogram and reliability diagrams.

4. A Table summarizing all the metrics you use (and the range of their possible values) would help. Added a table to the beginning of Section 6

**1.1.1 Minor comments and typos**

- Line 24: "has developed" $\rightarrow$ "have developed"?

  Suggested change made

- Line 38: 9km a.s.l.

  We have added a statement in the text that all heights are in reference to mean sea level.

- Line 68: In addition to these physical mechanisms, could dilution below detection threshold explain also part of this decrease?

  Yes this is what we meant by "dissipation due to dispersion". We have changed the wording to "the decrease in detected mass over time is due to ..... dilution below detection limits..."

- Line 100: Line 103: "This is expected to produce a better forecast than for instance using only one cycle"...this is to be checked. For example, by mixing several forecast cycles you may introduce inconsistency in the wind fields, something undesirable in Eulerian frameworks (may be not that bad for HYPSPLIT).

  Changed to For the HYSPLIT runs, the most recent forecast was always utilized, e.g. for a 12 hour forecast two or three forecast cycles would be utilized. This does not faithfully represent an operational framework in which future forecast cycles would obviously not be available. We expect it to produce a better forecast than for instance using only one cycle because of the overall increased accuracy in wind speeds and directions. However this has not been investigated in depth and some inconsistency in the wind fields may be introduced which could result in degraded model performance.

- Line 111: a mass fraction of ash of 0.1?

  Yes. We added in the value of 0.1 which was missing from the sentence.

- Line 187: "right" → "left"?

  We have changed the wording to make this more clear. The large shift of the CDF to the left seen in Figure 6b is due to the multiplicative factor. Then there is a small shift to the right caused by the addition of a constant which in this case is 2.2. This is apparent if you notice that the lowest values in the CDF have changed from near 0 to near 2.2.

- Line 194: 0:00Z → 0:00 UTC

  Changed Z to UTC

- Line 197: "This indicates a possible issue with turbulence parameterization in the model which control the rate at which the plume disperses". This is true, but actually could be more complex as diffusion effects can actually mix with wind shear advecting differently as particles settle down. With passive (non vertically resolved) satellite observations it is impossible to distinguish the single contributions from these two effects.

  We agree that it is difficult to say for certain with the current evidence what the reason is for the observed mismatch in modeled and observed plume dispersion. Therefore we removed this first sentence sentence of the paragraph which says "This indicates a possible issue with turbulence parameterisations in the model which control the rate at which the ash cloud disperses."

- Line 266, eq (1). Please make evident that C depends also on time.

  We have changed this to C(t) to indicate C is a function of t.

- Line 275: "information about the relationship between concentrations in adjacent grid cells for each member is not preserved". I do not understand this sentence. Do you mean that HYSPLIT does not output concentration at height levels at different periods?

  HYSPLIT does output a full 3D concentration grid. We meant that information about the relationship between concentrations in adjacent grid cells for each member is not preserved in the APL or ATL field. We have modified the statements to say that.

- Lines 275-285. Argument difficult to follow, please explain better. Why dosage cannot be computed individually for each ensemble member and then do your percentiles?

  It can and we stated this in Line 268 colorred now Line "For probability of exceeding a critical dosage $D_C$, this equation should be applied to each ensemble member and ensemble relative frequency of exceeding the dosage computed from the resulting ensemble of dosages".

The conversation here was meant to convey that if an end user does not have access to the full ensemble data, and only has APL or ATL data, then information on dosages cannot be accurately inferred from those fields. We edited the paragraph to make this more clear.

- Line 310: "P(..), giving the probability of observing. . ."

  Suggestion adopted.

- Line 359: (b)?

  Changed to Figure 11 (b)

- Line 395: You can also cite Folch et al. (2022) here, where skill scores are generalized to probabilistic contexts.

  The citation here is specifically for the NEP which as far as we can tell was first used by Ma et. al. We do cite Folch et. al. (2022) in Section 6.7 where the contingency table statistics are discussed.

- Line 406: Figures → Figure

  corrected

**1.2  Reviewer 2**

- Line 1. "Satellite retrievals of column mass loading of volcanic ash are incorporated into the HYSPLIT transport and dispersion modeling system". Does that mean that the procedures illustrated in the paper are now available with Hysplit?

  We aren't quite sure what you mean by "available with HYSPLIT". All of this work is repeatable with code available through the HYSPLIT distribution as well as code and scripts in publicly available github repositories for pre and post processing. The data used is also available upon request.

- Line 7. "the end of life of the ash cloud". What do you mean with "end of life". Do you refer to the settling of ash particles on the ground, or do you refer to ash concentration in the atmosphere getting lower than a fixed threshold?

  Added "when only small areas are still detectable in satellite imagery"

- Line 10. "small pieces of ash". What are the small pieces of ash? It is not clear if small refer to the size of tephra, or to small portion of the original ash cloud. If you are referring to a portion of the ash cloud, please change to "parts", also in the rest of the text, "pieces" is confusing.

  Changed "small pieces of ash" to "small areas of ash" throughout the text.

- Line 20. "However, within the next five years". Please add a reference.

  We added a reference.

- Line 38. "The resulting ash cloud reached a plume height of around 9km" Whenever a plume height is given, it should be stated if it is above sea level or above the vent. Fruthermore, according with Horvath et al. 2021 (https://doi.org/10.5194/acp-21-12207-2021), "At 20:50 UTC (Fig. 10f and g), the overshooting top of the eruption column reaches its maximum altitude of 15.3 km according to the side view technique".

  Thank you for the reference. We added a statement that all heights in the text are in reference to mean sea level. We also added Horvath to citation list and modified discussion here to include their results.

  We also performed some inversions allowing heights up to 16.88 km and added a paragraph *A few inversions were performed with heights up to 16.88 km as Horvath et. al. 2021 found that the maximum plume height reached 15-16 km. However results from those runs estimated no significant emissions above 12 km and we do not discuss them further here. This is not surprising as most of the mass is concentrated at the umbrella cloud height rather than the top height and these may differ significantly (Matin et. al. 2009).*

- Lines 42-44. "The resulting plume forms a complicated three dimensional structure as it is stretched and folded by the wind field over the course of less than a day. As shown later, the exact location and shape of these structures is difficult to forecast." Throughout the manuscript, the use of the terms "plume" and "cloud" are sometime confusing. For example, in line 38, the term plume seems to be used for the volcanic column. Because of that, here it is not clear if the 3D structure of the plume refers to the column or the cloud. I think that a choice should be done at the beginning and the use of the terms should be clearly stated, and then it should be consistent throughout the paper

  We changed most instances of plume to either eruption column, cloud, or ash cloud.

- Line 46. "most of the ash is drawn into an area of low pressure, the location of which is fairly easy to forecast. The end fate of these large ash clouds is similar to that of the Bezymianny cloud discussed here". Here the text seems to suggest that a pressure gradient drives the ash trajectories, while it is the drag exerted by the wind that controls them.

  We have changed "*drawn into*" to "*drawn toward and around*" to reflect that wind direction is not perpendicular to pressure gradient due to rotating frame of reference.

- Line 49. "path of ash parcels emerging from a low pressure area". I don't understand what this means. How do ash particles emerge from low pressure areas?

  We have shortened this sentence to simply read *location of the smaller structures can be even more difficult to forecast because of the time passed.*

- Line 50. Before this paragraph, I would be useful and interesting to have more details on the event (VEI, mass eruption rate, total grain size distribution, observed concentrations, deposit distance, wind condition at the vent), in order to have an idea of the size of the eruption.

  We added a reference to Appendix A which briefly dicusses meteorological conditions to this section. We also give wind speed at 350 mb.

  As far as we know there is no information on the total grain size distribution, observed concentrations, or deposit distance for this event. We don't feel trying to calculate a VEI is useful. MER is discussed later in the text as it is something we estimate. Providing it earlier may be confusing.

  Line 52. "the dissipation of the ash cloud". Maybe "dispersion" instead of "dissipation". I've never seen this term used for ash cloud by the volcanological community.

  We agree that it is not widely used but feel the word is appropriate here. We use dissipation to indicate we mean the actual disappearance or loss of the ash cloud. We have added the following to the sentence - *predict the dissipation of the ash cloud by which we mean its gradual disappearance.*

  Lines 56-61. It is important to give more information about the satellite retrievals. It is difficult to understand how the total mass or the total area are computed without knowing what are the data used. What is the maximum/minimum size of particles detected? Are you using an estimated columnar content? Is there a threshold used for the detection? Is there an uncertainty associated with the detection? To better understand how the area is computed, it would also be useful to have here two satellite images at different times, with a contour delimiting the area.

  A paragraph with these details on the satellite retrievals has been added to Section 3. A few more references were added as well. We describe how the area is computed but in an effort to keep the length of the paper reasonable, we did not add a graphic as requested. Various other figures throughout the paper show the mass loading field and we didn't feel that adding another one was necessary.

- Line 64. "as well as a mass fraction of fine ash of 0.1". Please justify this value and add a reference. Line 64. "this would result in a plume height of about 8.2 km". Without knowing the wind condition, it is difficult to judge if the use of the Mastin 2009 relationship gives reliable results. Is the plume strong or weak?

  The calculations performed with the empirical equation from Mastin 2009 is simply for comparison because such values are widely used as first guesses, in operational settings, and when only

plume height information is available. Further analysis of whether the relationship is applicable and whether the plume is strong or weak is not within the scope of the paper.

Since we also discuss this relationship in section 4.2, we have removed the discussion of it from section 3. In section 4.2 we changed the wording to make clear that the values calculated from the Mastin 2009 are only for a comparison. We also add in a reference for the mass fraction of fine ash and give the possible range for such values.

- Line 68. "the decrease in mass over time is due to physical processes such as dissipation due to dispersion". Mass is not dissipated, it is always conserved. The local concentration could decrease because of dispersion, but this does not decrease the total mass.

  Changed wording to *"the decrease in detected mass over time is due to ….. dilution below detection limits"*

- Line 69. "gravitational settling, and wet and dry deposition". Here it seems that settling and deposition are two different things. Is this correct?

  Yes. In short, gravitational settling accounts for the downward movement of particles due to gravity while deposition accounts for removal of mass from the atmosphere.

- Lines 80-84. This paragraph needs more details on the way Hysplit was used. From my knowledge, Hysplit can be used with particles or puff and, accordingly with this choice, with different dispersion/diffusion model associated with turbulence. Also, the way ash concentration is computed depends on the choice of particles or puffs. In both the cases, have you performed a convergence analysis on the number of particles/puffs needed to obtain a stable output? Furthermore, being Hysplit a Lagrangian model (i.e. it does not solve directly for mass concentration), it should be explained how concentration is computed from particles/puff position.

  HYSPLIT was run as a particle model. Yes, we did perform some analysis to make sure particle number was adequate. The requested details were added to section 4 and an additional appendix, Appedix C.

- Line 110. "plume width was 1 km". Again, it is not clear here if with "plume" you are referring to the volcanic column or to the ash cloud. Please clarify. If you are referring to the volcanic column, it is not clear where you assume this width/diameter? Is it at the base or at the top? In general, the diameter grows a lot from the base to the neutral buoyancy level. It is also not clear how the puffs/particles are released for this RunA. Is it from a cylinder, from a line, from the lateral surface of a cylinder? Maybe this does not make any difference, but in any case, if the reader wants to try to replicate the results, he/she needs to have all the information required.

  – We changed "plume height and plume width" to "eruption column height" and "eruption column width".
  – We changed the description from
    *Plume top height was 12.88 km and plume width was 1 km. Vertical mass distribution was uniform from the vent at 2.88 km asml to the plume height.*
    to
    *Initial mass distribution was uniform throughout a cylinder centered at the vent with width 1 km. The base of the cyclinder was at 2.88 km, the vent height. The top of the cylinder was at 12.88 km.*
  – We appreciate that the work should be reproducible and hope the revised wording is more clear. We would like to note that we have provided CONTROL and SETUP files for RunA and other runs in the github repository. These are the input files for HYSPLIT and contain all the information needed to reproduce runs.

- Line 110. "Vertical mass distribution was uniform from the vent at 2.88 km to the plume height". Is the mass distribution or the release of ash uniform? In both the cases, is this assumption justified? For small ash particles, most of the mass reach the neutral buoyancy level and it is released in the atmosphere at that height.

- – The emission rate remains constant for this run throughout the 2 h emission duration. To better communicate this we changed
  *"Initially the mass eruption rate was set at"*
  to
  *"Initially the constant emission rate was set at".*
  The initial distribution of mass is also uniform as described.

- – These assumptions are justified because this is a control run which is supposed to be similar to an operational setup at a VAAC. We added a sentence specifying that RunA is a control run at the beginning of the section. We also added *"This is similar to default operational settings at Washington and some other VAACs (Witham et. al., 2007, Beckettt et al., 2020). A uniform vertical mass distribution is not particularly realistic, but it is practical for current operations as it provides information on movement of ash from all heights that can then be further interpreted by the analyst issuing the final forecast.*

- Line 111. "and a mass fraction of fine ash" Please specify the value used.

  We added the value of 0.1 which had been left out.

- Lines 112-114. "would result in an MER of . . . ". It is not clear if the MER refers to the eruption rate of fine ash only or if it is the total rate. In general, this term is used to the total rate, so a different use would be confusing. If it is already the total eruption rate, the values reported in these lines seems low for a 10km volcanic column.

  Thank you for pointing out that we sometimes used MER when we meant MER of fine ash only. We have adopted the use of $MER_f$ for when we mean MER of fine ash.

- Line 116. "The inversion algorithm". Because Hysplit allows for inversion of trajectories, I think that a reader could get easily confused here. Please give some more details on the inversion procedure from Chai et al.

  We added a short description to make clear that the inversion algorithm utilizes forward HYS-PLIT dispersion runs.

- Line 118. "and 1 km in the vertical and area of above the vent" Something is missing here.

  We didn't properly describe the volume each run represented. We have changed this to: *Each individual HYSPLIT run used in constructing the TCM released approximately 20,000 particles representing 1 unit mass over one hour over a cylindrical volume 1 km in the vertical and 1 km diameter centered at the vent.*

- Line 118. "The modeling system only consider ash passively advected by the wind". Aren't you considering the gravitational settling?

  We took out that sentence and changed the following to *The modeling system does not consider plume dynamics and spreading of the umbrella cloud is only taken into account by utilizing an intial source term above the vent with some width.*

  Line 128. "with the mass distributed" Please change "mass" to "mass release".

  We made the requested change.

- Line 143. "The difference between using 20 and 6 $\mu$m particles". What is the difference in the settling velocity for the two sizes?

  Settling velocity for the 20 $\mu$m particles is about 0.03 m s$-1$ and settling velocity for 6 $\mu$m particles is about ten times smaller.

- Line 148. "lower emissions" Does "lower" refer to the height or to the rate?

  We have changed this to *"smaller emission rates"*

  Line 153. "we find that this is in large part due to the dispersion of the ash cloud not being adequately represented by the model" Is this really a limitation of the model or a limitation or of the retrieval algorithm?

  We don't have any reason to believe it is a limitation of the retrieval algorithm.

- Line 162. "Bias correction with CDF matching" Starting from this section, I really struggled to understand what has been done, mostly because the algorithms and techniques applied are not described with enough details to make then clear. For example, for the CFD matching, it is written that "model values and observed values" are sorted, but there is no mention to what are the values used. Are they probabilities, concentrations, pixel values, values averaged for all the pixels? I really have no idea. And the figure does not help, because there are no units on the x-axis. You need first to explain clearly for which parameter/variable you compute the CDF

  We have made the description of the method in Section 5 more detailed including specifying that we use the mass loading field. The axis labels in Figure 6 were modified to be more descriptive and include units. We also added more references.

- Line 167. "A linear fit is applied to the difference between the pairs as a function of forecast value. Although, Reichle and Koster (2004) use higher order fits, we find that a linear fit is adequate. " Maybe I looked at the wrong reference, but I can't find any mention of fit in the paper cited here. Please check.

  We have added more references and changed this to: *"A linear fit is applied to the difference between the pairs as a function of forecast value. Higher order fits or other functional forms can be used or a non-parametric approach sometimes referred to as quantile mapping can be used (Piani et. al. 2010, Gudmundsoon2012)*

  Our original statement is somewhat inaccurate as it looks like Reichle and Koster (2004) actually utilize the non-parametric approach.

- Line 172. "among the number of ash layers present". What do you mean with layers of ash?

  We changed this to *"among the number of vertical levels containing ash."* We also added some equations and more description that should make this more clear.

- Line 172. "mass loading values". Usually the volcanological community use this for the deposit (i.e. loading on the ground).

  In section 3 we added a comment that the atmospheric column mass loading is hereafter referred to only as mass loading. The term mass loading is widely used in similar work to mean the atmospheric column mass loading. As this paper does not discuss deposit properties at all, this should not be confusing.

- Lines 187-190. I would move these lines in the previous subsection.

  These lines occur in Section 5.1. We added the line "As these procedures can decrease and increase the spread of the forecast cloud respectively, the intercept can be loosely interpreted as an indicator of how well the spread of the forecast cloud matches that of the observed" to the previous section which is Section 5.

- Line 195. "Increasing the spread of the cloud at early times to more closely match observations, caused the spread of the cloud at later times to have a larger mismatch." The umbrella cloud intrudes as a gravity current at it takes some time to reach the maximum upwind and crosswind spreading. Have you tried to increase the size with time?

  Modeling the plume dynamics is beyond the scope of this study.

  The two simple cases we looked at could be considered bounding cases. In one case the initial spread was small for the entire eruption period. In the other case the initial spread was large for the entire eruption period. Creating runs in which the initial spread started out small and then grew to be large would presumably result in a modeled cloud somewhere between those two cases. The conclusion that increasing the spread of the initial plume tends to increase the spread of the cloud at later times would probably remain the same.

- Line 198. "The modeled horizontal dispersion of the cloud is too fast ". This could be also done to a release of ash particles from a too large vertical interval, coupled with vertical wind shear. What happens when a larger fraction of particles is released from the neutral buoyancy level? As previously commented, particles of size you are considering here should, in large part, reach the top of the column.

As discussed in Section 5.2, the same trend in the values of the intercepts is seen with the use of the emissions estimated from the inversion algorithm. This source term does release most of the mass around the neutral buoyancy level. We have emphasized this point by adding the sentence *This indicates that even with the improved source term, the modeled cloud is initially more compact than observed and then becomes more dispersed than observed.* to Section 5.2.

We also removed the first sentence sentence of this paragraph which says *"This indicates a possible issue with turbulence parameterisations in the model which control the rate at which the ash cloud disperses."*

- Line 215. "We make the assumption that verification of modeled column mass loading values can be used as a proxy for verification of forecast concentrations." The distinction between the use of column of mass loading and concentration should have been done at the beginning, because in most of the previous sections it was not clear which of the two forecasts and observations were referring to.

  To Section 3, we added a better description of the fields provided by the satellite retrievals and indicate that it is the column mass loading which is utilized throughout the paper.

  To section 5, we specify that it is the column mass loading was being used for the CDF matching.

- Line 229. "output of ash column mass loadings shown in Figure 8" The use of a linear color scale makes more difficult to compare the results. Please use a log scale. Also the choice of colors does not help.

  See response in Section 1.2.1.

- Lines 233-234. "By 12 UTC, this line of ash has broken into three small pieces, one just to the east of the volcano, one to the south, and one to the northwest."

  Are these parts at different heights?

  Yes they are. We have modified the sentence to include the retrieved top heights. *By 12 UTC, this line of ash has broken into three small areas, one just to the east of the volcano which has the highest retrieved top heights of around 10-11 km, one to the south with the lowest retrieved top heights of between 5-7 km, and one to the northwest with retrieved top heights of between 7-9 km. Presumably there might be ash at very low concentrations still connecting the pieces.*

- Line 249. "An example is shown in Figure 9(c) which shows number of model runs exceeding a given mass loading threshold" I would remove "an example", because the number is not normalized in figure 9(c).

  Removed those words.

- Line 250. "84 %". I think that here and in the rest of the text you should remove the space between the number and the percentage symbol.

  The space is in accordance with ACP style guidelines. See https://www.atmospheric-chemistry-and-physics.net/submission.html#math

- Lines 259-261. "In later sections we will utilize measures such as the precision recall curve, PRC, to evaluate the ATL at various probability thresholds. Keep in mind that these statistics are the same for the APL with the caveat that the point for 5 This is difficult to understand, because the PRC has not been introduced so far in the manuscript.

  We have removed this line from this section. Instead we refer back to this section and Figure 9 when discussing the PRC curve.

- Line 263. "Some sources" Which sources? Add references.

  Added references

- Line 267. "If velocity is constant then D is not sensitive to spatial averaging that is performed parallel to the flight path." This is true, but when computing the concentration in the grid cell,

averaging is preformed both in directions parallel and normal to the flight path, so computed dosage depends on grid resolution.

We agree that is true and don't see a contradiction with our statement.

- Line 292. "Instead predicting the time at which the ash cloud breaks into small enough pieces to be ignored becomes important." Written in this way it seems that it is common to observe the ash cloud breaking into small parts, but I'm not so sure it is always the case. It also not completely clear to me what "breaks into small pieces" means. Is it because between these "small pieces" ash concentration is very low and so it is not detected? Or is the ash cloud really splitting into different and isolated parts?

  We have changed this to say *"Instead predicting the time at which the ash cloud is small enough to no longer be of concern becomes important."*

- Lines 295-298. "The extent of lower mass loading of ash, $0.2$ g m$^2$ continues to increase through 12 UTC. In contrast, the extent of the higher mass loadings follows the observations more closely. This mismatch occurs because the spatial gradient in the observations is much steeper than in the simulation. This is again an indication that the modeled turbulent dispersion which controls the spread of the ash is not reproducing what is observed." This is not clear to me. If the extent of the area exceeding a threshold is too large, that does not necessarily mean that there is problem with turbulent dispersion. In fact, when the bias correction is applied, it seems that the areas are better reproduced.

  We have changed this to the following (changes in bold) to highlight that the issue is a combination of several things; the extent of the higher mass loading area being modeled fairly well while the extent of the lower mass loading area is overpredicted fairly severely, along with the observation that the reason this occurs is the steeper spatial gradient of the mass loading field in the observations. We also softened the statement by saying "could be" rather than "is".

  *Without bias correction, both RunA and RunM overpredict the lifetime of the ash cloud significantly. The extent of lower mass loading of ash, $0.2$ g m$^{-2}$. continues to increase through 12 UTC* (**Figure 10(a) left column**). *In contrast, the extent of the higher mass loadings follows the observations more closely* (**Figure 10(a) right column**). *This mismatch occurs because the spatial gradient* **of the mass loading field** *is much steeper in the observations than in the simulation* **as can be seen in Figure 8**. *This* **could be** *an indication that the modeled turbulent dispersion which controls the spread of the ash is not reproducing what is observed*

  The bias correction does work fairly well, but it is still preferable to find the reason for the bias and correct it in the modeling if possible.

- Lines 309-315. I'm sorry but the description is the reliability diagram is extremely confusing to me. In the simulations you have the ensemble of simulations, the output at different times, the output at different pixels. When you write "the probability of observing the event" what do you mean? Is it the probability for a single element of the ensemble, considering all the pixels at a single time? Is it the probability associated with the variability in the ensemble elements? Is it the variability associated with the different output times in a time interval? I have the same problem with the variable plotted in the vertical axis of the middle column in Fig.11, what does "Fraction observations" mean? I have the same problem with the refinement distribution. "The second part of the diagram is the refinement distribution which is a histogram showing how often the modeled probability, yi, occurred". You need to state more clearly what you mean with "modeled probability".

  We have re-written the description of the reliability diagram, refinement distribution, and calibration function.

  We have also changed the x and y axis labels on the middle and right plots in Figures 11 and 12 which show the calibration function and refinement distribution.

- Section 6.5.2 In this section I have problems to understand the discussions on the reliability diagram and the refinement distribution because, as written above, it is not clear what is plotted in the figures.

  We have reorganized and reworded the discussion of the reliability diagram.

- Lines 379-380. "Observed and modeled fractions are then computed for different neighborhood sizes, n, by convolution of the field with a square kernel of that size". I suggest to change to: "Observed and modeled fractions are computed by convolution of the field with a square kernel, for different kernel sizes, n.".

  We have changed this to

  *Observed and modeled fractions are then computed for different neighborhood sizes, n. Here that is achieved by convolution of the field with a square kernel of size n.*

  We hope this reflects that the main idea is to compute observed and modeled fractions at different spatial resolutions. It is usually done by convolution with a square kernel, but a different type of kernel could be utilized as well.

- Line 384. Before equation 3, please define O(n) and m(n). I assume they are the fractions, but it would be better to state it explicitly.

  We have added descriptions of the variables in this equation after the equation.

- Lines 389-391. "When computing the FSS, it is standard for the reference forecast to be defined as the largest possible MSE that can be obtained from the forecast and observed fractions". Isn't this number just 1? By looking at Eq.3, you maximize the MSE when you maximize each addend of the sum, and this is obtained when, in the difference between O(n) and m(n), one value is 0 and the other is 1. I also don't understand why in Eq.5, with respect to Eq.3, the square moved inside the parenthesis and there is a difference instead of a sum.

$$\text{MSE}(n) = \frac{1}{N_x N_y} \sum_{i=1}^{N_x} \sum_{j=1}^{N_y} \left( (O(n)_{i,j} - m(n)_{i,j} \right)^2 \tag{5}$$

$$\text{MSE}(n) = \frac{1}{N_x N_y} \sum_{i=1}^{N_x} \sum_{j=1}^{N_y} O(n)_{i,j}^2 - 2O(n)_{i,j} m(n)_{i,j} + m(n)_{i,j}^2 \tag{6}$$

  For the reference forecast, the negative term $-2O(n)m(n)$ is dropped. Also $O(n)$ and $m(n)$ take on values between 0 and 1. They are not always 0 or 1.

- Line 397. "At some value of n ¡= 2N-1" What is N?

  We added a sentence *$N_x$ and $N_y$ are the number of grid cells in the x and y directions and N is used hereafter to refer to the larger of these.*

- Line 419. "Number of occasions" Here I have the same problem I had with the description of the reliability diagram. The definition of the parameters seems to ignore that you are applying the technique to a specific application, and it is given as an abstract definition. What does the term "occasions" mean? I don't understand if you refer to the number of pixel for a simulation of the ensemble, or to the number of ensemble for a pixel.

  We have added the following paragraph to this section. *The following steps are used to create contingency tables for the probabilistic forecasts. First the ATL field is computed for a given mass loading threshold. The ATL field consists of values from 0 to 100 %. To convert the ATL field to a binary field, a probability threshold is applied as shown in Figure 9(c). Alternatively one could start with the APL field shown in Figure 9(e) and then apply the mass loading threshold threshold to arrive at Figure 9(d).*

- Line 431Line 431. "For instance, a measure such as F is highly sensitive to the domain size " What is F here?

  This was meant to be POFD. The correction has been made.

- Line 435. "various probability thresholds". Is this correct or should it be just "various thresholds", without probability? Is this the same threshold used to define a in Lines 419-420? "both the forecast and observation are above threshold". If it is really a probability threshold, please can you write explicitly to what the probability refers to?

It is meant to be probability thresholds. We have added a paragraph to this section as described previously.

- Line 436-437. "However, some care must be taken in the interpretation of the curves as POFD which is plotted on the x axis". Where is the x-axis you mention? There is no reference to any figure here.

  We don't provide plots of ROC curves because we determined they were not a particularly good tool. Here we are just providing an explanation of why we think they are not good for evaluation. We have modified the sentence as follows (addition shown in bold).

  *However, some care must be taken in the interpretation of the curves as POFD which is plotted on the x axis of the* **ROC curve (not shown)***, is dependent on d and thus can be increased or decreased by changing the size of the domain.*

- Line 443-end of subsection. I confess that I was lost here, because I could not understand clearly the probability thresholds, as written above.

  We believe that the addition of the paragraph describing the process for creating the contingency table will make the paper much more readable.

- In the two first columns of Fig.16 there are also a lot of markers, but I can't understand what they represent. Probably there is something I missed.

  Some description was left out of the figure captions. We have added it in.

**1.2.1 Figures**

- Figure 1. Please add letters to the panels. In panel (a), is is $e^{(-xx)}$, with e being the Euler's number, or $10^{(-xx)}$? In Panel (d), please write if the height is a.s.l.

  - Added (a),(b),(c),(d)
  - The first panel is on a natural log scale. It is labeled correctly with e being Euler's number. To make this absolutely clear we have added to the caption **The y axis is a natural log scale** *and the orange line indicates a period of exponential decay*
  - We have added a statement to the paper that all heights are in reference to mean sea level.

- Figure 6. In panel b and d add a label to the vertical axis.

  - For (a) and (c), units were added to x and y axis labels.
  - For (b) and (d), x label was changed from 'values' to 'Mass loading (g m$^{-2}$)'.
  - For (b) and (d) CDF label was added to the y axis.
  - For (e) and (f) fontsize was increased on x and y labels and (UTC) was added to x label.

- Figure 8. The color scale does not help to distinguish low concentrations from the background. I think it would also work better a log scale.

  We modified the right most column to show three contours on a log scale. We also tweaked the color scale on the other two columns.

  We made the lower end of the color scale slightly more saturated so low concentrations could be distinguished more easily. However we found that distinguishing the very low concentrations from 'background' is not particularly helpful and doing that too much (as well as using a log scale) creates the impression that the extent of the modeled ash cloud is much larger than it actually is.

  We think the linear scale is appropriate for the range of values of interest described in Section 6.1, (0.2, 2, 5, and 10). If modeled concentrations below around 0.01 cannot be distinguished from the background of 0, then we think that is also appropriate.

- Figure 9. I think you should remark that the color scale here is not linear or log, but I think a mix of the two.

  Added *(e) Concentrations at APL of 84 %.* **Note that the color scale is designed to convey information relevant to the thresholds described in Section 6.1.**